# Electrocatalytic CO₂ reduction to ethylene in an acid-fed membrane electrode assembly at 10 A

Derong Chen [1,2,3,7], Jia Liu[2,7], Yijia Yuan [2,7], Xiaocang Han[2], Kun Zhang[2], Qikun Hu[2], Shuhe Han [4], Shibo Xi[5], Quan-Hong Yang [1,3] ✉ & Kian Ping Loh [2,6] ✉

Electrocatalytic $CO_2$ reduction reaction ($CO_2$RR) using membrane electrode assembly (MEA) systems requires complex regulation of protons, hydroxyls, carbonate ions and alkali-metal ions across both electrodes to efficiently produce multicarbon products. In acid-fed $CO_2$RR MEAs, excessive proton migration and accumulation on the catalyst surface suppress $CO_2$ adsorption and promote hydrogen evolution, leading to low Faradaic and energy efficiencies. Sluggish hydroxide transport further triggers carbonate precipitation, undermining system stability. Here we report an acid-fed membrane electrode assembly system for highly efficient $CO_2$RR by integrating hydrazone-linked covalent organic framework (COF) and catalyst on the anion-exchange membrane to enable bidirectional pathway for hydroxide and potassium ions diffusion, while enhancing transport of $CO_2$ to the catalyst surface. As a result, the scaled-up MEA operates at a full-cell voltage of ~4.5 V under a total current of 10 A (current density of 204 mA cm$^{-2}$), delivering a Faradaic efficiency of ~50% for $CO_2$-to-$C_2H_4$ conversion and maintaining stability for over 300 hours.

$CO_2$RR using renewable electricity holds promise for ethylene ($C_2H_4$) production[1]. The use of MEA systems has emerged as the most promising approach to both achieve high Faradaic efficiency (FE) and energy efficiency (EE) because of its low ohmic resistance and enhanced reaction kinetics compared to conventional flow-cell systems[2]. In MEAs, the close proximity of the cathode/ion exchange membrane interface and gas diffusion electrode (GDE) facilitates faster electron transfer and ion conduction. This enables much higher current operation than flow-cell systems with voluminous electrolyte, which are essential for improving the overall efficiency of $CO_2$

reduction reactions. Other advantages of the MEA systems include their ease of scalability compared to traditional flow reactors, and their modular design allows for easier integration into larger systems without significant losses in efficiency, making them commercially viable options for industrial applications aimed at $CO_2$ conversion.

Alkaline and neutral electrolytes have been employed in $CO_2$RR MEA systems. However, these configurations are challenged by bicarbonate formation due to the reaction of $CO_2$ with hydroxide ions (OH⁻) at the cathode. This parasitic reaction not only depletes $CO_2$ but also destabilizes system operation and reduces the MEA's operational

[1]Joint School of the National University of Singapore and Tianjin University, International Campus of Tianjin University, Binhai New City, Fuzhou, China. [2]Department of Chemistry, National University of Singapore, 3 Science Drive 3, Singapore, Singapore. [3]Nanoyang Group, Tianjin Key Laboratory of Advanced Carbon and Electrochemical Energy Storage, School of Chemical Engineering and Technology, and Collaborative Innovation Centre of Chemical Science and Engineering (Tianjin), Tianjin University, Tianjin, China. [4]Department of Applied Physics, The Hong Kong Polytechnic University, Hung Hom, Kowloon, Hong Kong, China. [5]Institute of Sustainability for Chemicals, Energy and Environment, A*STAR, Singapore, Singapore. [6]Centre for Hydrogen Innovations, National University of Singapore, E8, 1 Engineering Drive 3, Singapore, Singapore. [7]These authors contributed equally: Derong Chen, Jia Liu, Yijia Yuan. ✉e-mail: qhyangcn@tju.edu.cn; chmlohkp@nus.edu.sg

lifespan (Supplementary Fig. 1). To address these challenges, mitigation strategies such as periodic pulsing of water and solvents[3], adoption of bipolar membranes (BPMs)[4] or integrated anion exchange membrane (AEM)/proton exchange membrane (PEM) in MEA systems[5], and voltage cycling between operational and regenerative modes[6], have been developed to enhance system stability. However, some of these approaches require frequent activation processes and suffer from ionic conductivity issues or short-term stabilities owing to electrode flooding. Another deleterious effect is that carbonate ($CO_3^{2-}$) and bicarbonate ($HCO_3^-$) ions migrate through the AEM, acidified by protons ($H^+$), and converts to $CO_2$ on the anode side, leading to low carbon efficiency[7]. She. X et al. demonstrated that an alkali cathode MEA using pure water (alkali cation-free) could prevent carbonate precipitation, enabling sustained ethylene production (~50% FE at 1000 h)[8]. However, for a cell stack comprising six MEA cells, a high operation voltage is needed (25–27 V at 10 A), indicating the need for strategies to improve overall energy efficiency (EE) for $CO_2$ reduction.

In acidic electrolyzers, the abundance of $H^+$ converts locally formed carbonate anions back to $CO_2$ within the diffusion layer, mitigating $CO_2$ crossover and salt precipitation to some extent, thus offering a carbon-efficient platform for $CO_2RR$[7,9,10]. The main problem for $CO_2RR$ with acidic electrolyte is the occurrence of severe HER at elevated current densities. The performance of acid-fed flow cells suggests that the catalytic microenvironment plays an important part in the $CO_2$-to-$C_2H_4$ conversion. By establishing a high-concentration of alkali cations or anions in the catalyst microenvironment, significant enhancement in multicarbon production and overall system energy efficiency have been achieved[11–13].

The local concentration of ions and molecules plays an important role in determining the $CO_2RR$ performance in a zero-gap acidic MEA system. A strongly acidic electrolyte flows exclusively through the anode compartment, this setup creates competition between alkali cation ions and protons in the anolyte as they traverse the membrane to reach the cathode side[14]. The origin of low $FE_{C2H4}$ and $EE_{C2H4}$ in the zero-gap acid-fed $CO_2RR$ MEA electrolyzer can be traced to the transportation of protons from anode to cathode and their accumulation on the catalyst surface. This limits $CO_2$ adsorption and leads to severe HER, despite adding a high concentration of alkali cations in the acidic electrolyte. Such a scenario differs markedly from H-cells and flow cells, where the cathode catalyst operates in a relatively stable acidic environment. In addition, the sluggish migration of locally generated $OH^-$ from cathode to anode induces bicarbonate precipitation on the cathode during $CO_2$ reduction, despite the low pH of the bulk electrolyte, which hinders the sustainability of $CO_2$ reduction[15]. Consequently, the mechanisms governing cation effects in acidic $CO_2RR$ within MEA systems warrant further investigation, as the local microenvironment is complicated by the presence of various charged ions (e.g., $H^+$, $OH^-$, $K^+$, $HCO_3^-$) and molecules (e.g., $CO_2$ and $H_2O$)[16].

To address challenges facing MEA systems operating under acidic $CO_2RR$ conditions, we developed a strategy to selectively attract $K^+$ while repelling locally generated $OH^-$. This approach improves gas transport efficiency and sustains robust $C_2H_4$ selectivity. We used hydrazone-linked COFs functionalized with amine and ether groups, and coated it over the catalysts on the cathode to modulate the kinetics of $K^+$ and $OH^-$. This strategy enabled a high FEs of $C_2H_4$ of 54% and 82% for $C_{2+}$ operating at 600 mA cm$^{-2}$, with partial current densities of 311 mA cm$^{-2}$ and 490 mA cm$^{-2}$, respectively, when operating at pH-1 in the MEA cell. Finally, we demonstrated a scale-up MEA cell system with a full-cell voltage of ~4.5 V with a Faradaic efficiency of (50 ± 3) % for $CO_2$-to-$C_2H_4$ conversion at the current of 10 A (current density = 204 mA cm$^{-2}$) with long-term stability over 300 h.

## Results

### Analysis and the design of acid-fed MEA system

The use of the cation exchange membrane (CEM) is predicated upon its high cation conductivity and selectivity. However, acidic $CO_2RR$ using a CEM-MEA becomes unstable at high current densities. This instability arises from the sluggish migration of electrogenerated hydroxide ions (Fig. 1a), leading to bicarbonate precipitation in the cathode chamber and resulting in significant degradation of cell stability. The introduction of an anion exchange Membrane (AEM) in conjunction with the CEM creates a bipolar interface that aids in $CO_2$ regeneration for recycling, thereby partially alleviating salt precipitation issues. However, the continuous generation of gas molecules between the AEM and CEM membranes forms gas pockets that increase the Ohmic impedance and fluctuating overpotentials for reactions (Fig. 1b). This adversely affects the Faradaic and energy efficiency of ethylene production. To tackle these challenges, we propose revisiting the acid-fed MEA system utilizing an AEM, as depicted in Fig. 1c. The AEM facilitates the transport of $OH^-$ ions via an electromigration process to the anode, reducing the concentration of $OH^-$ near the cathode and enhancing stability by mitigating salt precipitation. While the Donnan exclusion effect of the AEM limits cation migration, $K^+$ and $H_3O^+$ ions may accumulate near the AEM due to potential drops at the catalyst-membrane interface. Driven by an electric field, these ions migrate from the AEM to the cathode, where they participate in $CO_2$ regeneration and conversion processes[19].

When using an acid-fed AEM-based system for $CO_2$-to-$C_2H_4$ conversion, the high HER ($FE_{H_2} > 30\%$) resulted in a low $C_2H_4$ selectivity ($FE_{C_2H_4}$, ~ 25%), this is despite the fact that a $K^+$ concentration of 1 M was added in the bulk anolyte to suppress HER (Supplementary Fig. 2). Increasing the interfacial $K^+$ concentration by pre-spraying electrode with potassium salt can enhance $C_2H_4$ selectivity, prolonged operation however led to salt precipitation (details in "Methods", Supplementary Fig. 3). These findings underscore the critical interplay between localized $K^+$ and $OH^-$ concentrations in the acidic $CO_2RR$ MEA system.

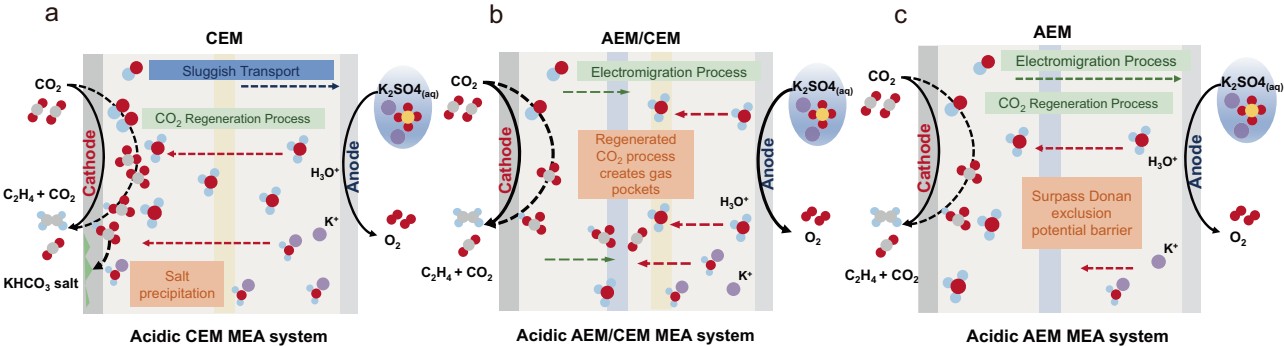

**Fig. 1 | Design of various acidic MEA systems. a** Acidic CEM MEA (**b**) Acidic AEM/CEM MEA system (**c**) Acidic AEM-based MEA system.

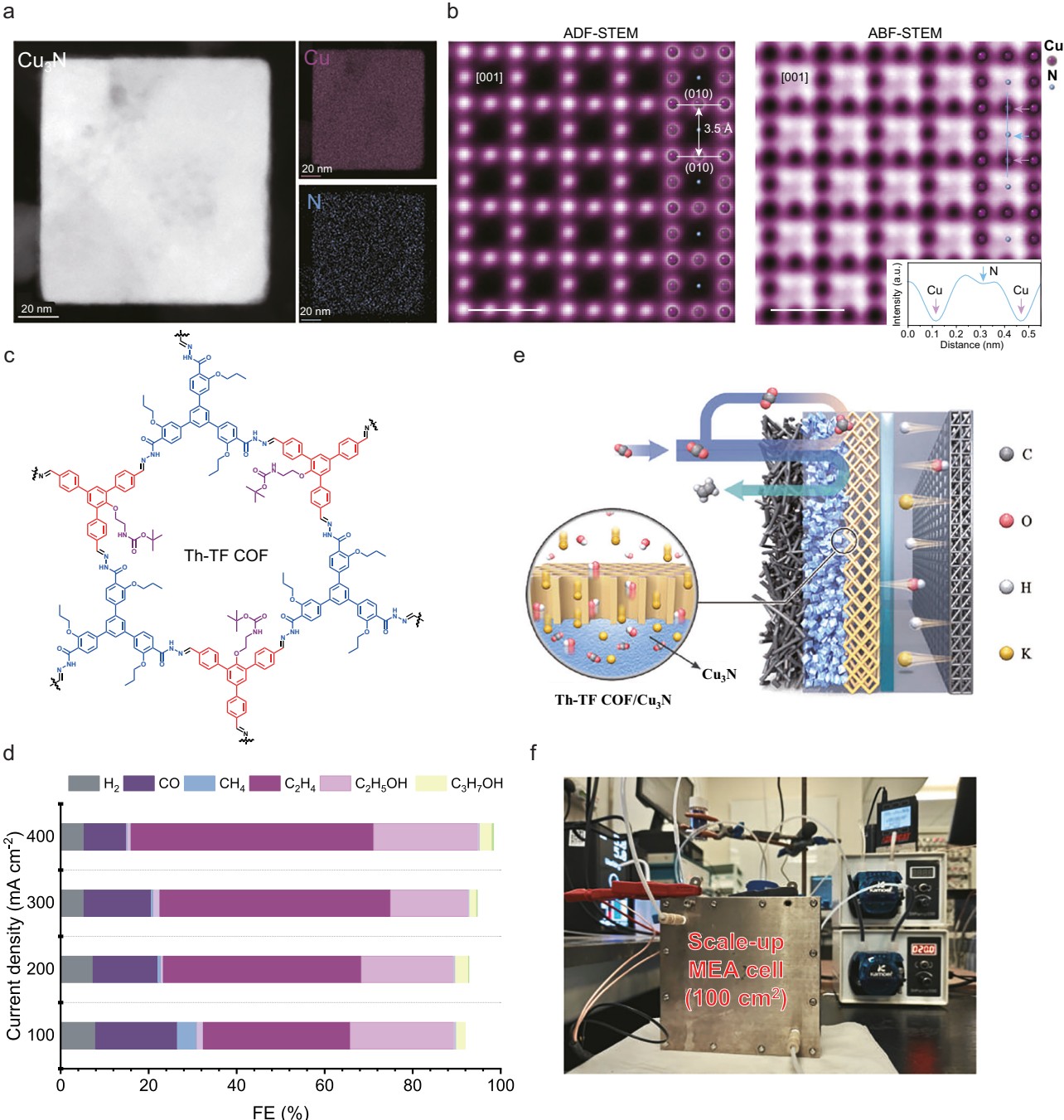

**Fig. 2 | Characterization of the Th-TF COF modified Cu₃N catalyst. a** EDS mapping of a pristine Cu₃N catalyst. Scale bars: 20 nm. **b** atomic-resolution ADF-STEM and ABF-STEM images of Cu₃N reflecting its lattice arrangement. Scale bars: 5 Å. **c** Schematic showing structure of the Th-TF COF and (**d**) $CO_2R$ product distribution in the acid-fed COF AEM MEA system at current density ranging from 100 to 400 mA cm⁻² in 0.5 M $K_2SO_4/H_2SO_4$ (pH-2) conditions. No iR correction was applied. Source data for Fig. 2d are provided as a Source Data file. **e** Schematic of the zero-gap COF-based acid-fed $CO_2RR$ MEA system. **f** Photograph of scale-up electrolyzer cell (a total electrode area of 100 cm²).

Optimizing this balance is vital for sustaining high $C_{2+}$ product selectivity while avoiding parasitic HER and salt deposition.

## Synthesis and characterization of the Th-TF COF modified Cu₃N catalyst

To achieve a high $C_{2+}$ product in an acid-fed $CO_2RR$ AEM MEA system, we constructed a hierarchical layer-by-layer electrode configuration consisting of Cu₃N/COFs/anion-conducting ionomer (Sustainion), which is abbreviated as CNCP; see details in methods. Copper nitrite (Cu₃N) catalyst is selected for its high $CO_2$-to-$C_2H_4$ conversion[17–19]. The

Cu₃N nanoparticles were synthesized via a one-step pyrolysis method[17]. Powder X-ray diffraction (PXRD) was used to prove the pure phase of Cu₃N (Supplementary Fig. 4). Annular dark-field scanning transmission electron microscopy (ADF-STEM) revealed the cubic morphology of the pristine Cu₃N catalyst. Energy-dispersive X-ray spectroscopy (EDS) mapping confirmed the uniform distribution of Cu and N across the particle (Fig. 2a, Supplementary Fig. 5). The atomic-scale ordering of Cu and N was further resolved using atomic-resolution ADF-STEM and annular bright-field (ABF) STEM imaging (Fig. 2b). Due to the atomic number (Z)-contrast nature of ADF

imaging, the heavier Cu atoms appeared bright, while ABF-STEM—sensitive to light elements—clearly resolved the N sublattice as light dark contrast, supported by the corresponding intensity line profile (Fig. 2b). The observed lattice fringes with an interplanar spacing of 3.5 Å align with the (010) plane of $Cu_3N$, reflecting its characteristic lattice arrangement. The valence state and coordination environment of $Cu_3N$ were examined by X-ray absorption spectroscopy (XAS) and X-ray photoelectron spectroscopy (XPS). XPS of Cu atom in $Cu_3N$ catalyst showed spin orbit coupling-split peaks at 952.3 eV and 932.5 eV in the $Cu_{2p}$ spectrum (Supplementary Fig. 6a, b), which can be attributed to positively charged Cu species ($Cu^{1+}$)[17]. In addition, XPS of nonmetal elements was also used to probe the chemical state of nitrogen atoms. For $Cu_3N$, the peak of covalent N was detected at a binding energy of 397.3 eV (Supplementary Fig. 6c). Similar to the results of XPS, the Cu $K$-edge X-ray absorption near-edge structure (XANES) spectrum of $Cu_3N$ was collected to reveal its coordinated state. Extended X-ray absorption fine structure (EXAFS) spectroscopy confirmed the existence of the Cu-N bonding located at 1.34 Å in $Cu_3N$ catalyst, and the Cu-Cu coordination at 2.33 Å was observed in the Fourier transform-EXAFS spectra in Supplementary Fig. 7. These results indicate that the formation of phase-pure $Cu_3N$ catalyst.

Zhao Y. et al. utilized amphoteric COF and perfluorinated sulfonic acid ionomer (PFSA) to establish a proton-blocking, $K^+$-enriched environment and achieved high efficiency for multicarbon products (Faradaic efficiency; $FE_{C2+}$ ~ 75%; $FE_{C2H4}$ ~ 40%) in a flow system[11]. To achieve better performance suited to an acid-fed MEA system, we synthesized and screened acid-resistant COF with tailored reticular backbones and functional groups (Supplementary Fig. 8). We identified a triformylbenzene-derived COF (denoted as Th-TF COF) as the top performer (Fig. 2c). Th-TF COF has a hydrazone-linked ($R_1R_2C = N$-NH-$R_3$) framework, functionalized with an amine group and oxygen-containing alkyl chains. The synthetic procedures for monomers are provided in Methods and Supplementary Fig. 9. Ex-situ PXRD and FTIR confirmed the formation of the hydrazone framework, revealing a (100) reflection at $2\theta \approx 3.67°$ and fingerprint C = N, C = O, and N−H vibrations with red-shifts relative to the precursors (Supplementary Fig. 10). We found that Th-TF COF could effectively regulate local ion concentration ($K^+$ and $OH^-$) and interaction with $CO_2$ molecules, thereby mitigating competitive proton transport and carbonate precipitation in the acidic MEA system. The $Cu_3N$ catalyst was coated with Th-TF COF and loaded onto the gas diffusion layer (GDL) to be used as CNCP electrodes in the acid-fed MEA system, achieving a FE of 55% for $C_2H_4$ and 75% for $C_{2+}$ products at 400 mA cm$^{-2}$ (Fig. 2d), rivaling the highest reported values under similar pH conditions[7,9,11,12,20,21]. When integrated with an AEM acid-fed MEA system, the Th-TF COF facilitated migration and adsorption of local anions ($OH^-$) and cations ($K^+$), critical for stabilizing $CO_2$ activation and reduction pathways (Fig. 2e).

## Efficient $C_2H_4$ electrosynthesis in the acid-fed Th-TF COF AEM MEA system

We assessed the $CO_2RR$ performance of the CNCP electrodes in the acid-fed AEM MEA electrolyzer setup with 0.5 M $K_2SO_4$/$H_2SO_4$ (pH ~ 1) anolyte and an iridium oxide as the anode. In Fig. 3a and Supplementary Fig. 11, over the full current density range (100-600 mA cm$^{-2}$), the use of Th-TF COF greatly suppressed the competing HER (<20%) and improved $C_2H_4$ selectivity. At 600 mA cm$^{-2}$, the $FE_{C2H4}$ and $FE_{C2+}$ (53% towards $C_2H_4$, 25% towards $C_2H_5OH$ and 4% towards $C_3H_7OH$) values showed peaks of $(52 \pm 2)$ % and $(80 \pm 2)$ %, respectively, with a $C_2H_4$ partial current density ($j_{C2H4}$) of 311 mA cm$^{-2}$ and a $C_{2+}$ partial current density ($j_{C2+}$) of 490 mA cm$^{-2}$. The CNCP electrode achieved a peak of $EE_{C2H4}$ of ~20% with low full-cell voltage of ~3.5 V at 400 mA cm$^{-2}$ (without iR; i, current; R, resistance compensation; Supplementary Fig. 12). In contrast, the highest $C_2H_4$ selectivity of $Cu_3N$ electrode (Sus/$Cu_3N$) is limited to $(36 \pm 4)$ % with a low $C_2H_4$ partial current density of $(64 \pm 5)$ % at 200 mA cm$^{-2}$. The FE of $C_2H_4$ then dropped

dramatically when current densities increased, the $C_2H_4$ FE is only 15% at 400 mA cm$^{-2}$ due to the severe HER ($FE_{H2}$: ~60%, Supplementary Fig. 13). The value of $FE_{C2H4}$ of CNCP electrode remained nearly constant as the electrolyte acidity was reduced from pH 1.2 to 6.3 (Supplementary Fig. 14). This evidences that the CNCP electrode could maintain high catalyst activity even with changing surface pH during high current densities $CO_2$ reduction.

The superior $CO_2$-to-$C_2H_4$ conversion performance of CNCP in the acid-fed AEM MEA system prompted us to scale the MEA cell to operate at 10 A in a 100 cm$^{-2}$ electrolyzer MEA cell (Fig. 2f), where both $C_2H_4$ selectivity ($FE_{C2H4}$) and specific productivity ($SPCE_{C2H4}$) in acidic $CO_2RR$ were strongly enhanced. As shown in Fig. 3b, c, the CNCP electrode achieved a $C_2H_4$ faradaic efficiency (FE) of ~53% with a total $C_2H_4$ production current of 5.5 A (current density: 204 mA cm$^{-2}$) across a 49 cm$^2$ reaction area. Single-pass carbon efficiency (SPCE) measurements revealed a peak $CO_2$-to-$C_2H_4$ utilization of 16% at 10 A when reducing the $CO_2$ feedstock flow rate to 60–100 sccm (Fig. 3d), ranking among the highest reported values for acidic $CO_2RR$ scale-up MEA systems. The acid-fed AEM MEA system demonstrated stable operation for over 300 h at 10 A with a full-cell voltage of 4.2–5.0 V (no iR compensation) and an average $C_2H_4$ production rate of 25.91 mmol h$^{-1}$, with the ~50% Faradaic efficiency benchmark sustained during the first ~100 h (Fig. 2e, Supplementary Table 1). In contrast, a Sustainion coated-$Cu_3N$ (Sus/$Cu_3N$) AEM MEA system with 0.5 M $K_2SO_4$/$H_2SO_4$ anolyte showed unstable performance within 30 h due to dominant hydrogen evolution ($FE_{H2}$: ~60%) over C−C coupling (Supplementary Fig. 15), underscoring the critical role of Th-TF COF in stabilizing ion transport and interfacial reactions.

The enhanced $CO_2$-to-$C_2H_4$ performance stems from the synergistic effects of the CNCP architecture and Th-TF COF's ion-regulation capabilities. Post-electrolysis SEM analysis confirmed the preserved microstructure of the catalyst and Th-TF COF (Supplementary Fig. 16). To elucidate the origin of the gradual performance decay, we conducted a series of post-reaction characterizations. XRD and operando XAFS reveal that the characteristic reflections of $Cu_3N$ disappear after extended operation, while the Cu K-edge XANES confirm a phase transformation into $Cu^0$. In contrast, Raman spectra show that the characteristic vibrational bands of the Th-TF COF remain unchanged after 300 h, thereby excluding oxidative degradation of the framework. Complementary TEM/EDS and cross-sectional SEM-EDS analyses demonstrate progressive nitrogen depletion of $Cu_3N$ and electrolyte infiltration into the GDL, which together account for the steady decline in FE and cell voltage (Supplementary Figs. 17, 18). In addition, gradual flooding of hydrophobic domains by liquid $CO_2$ reduction products (e.g., $C_2H_5OH$), which increases electrolyte penetration, likely contributed to performance decline after 300 h. Further optimization of gas diffusion electrodes (GDEs) will be essential to improve long-term stability. Overall, the Th-TF COF-modified $Cu_3N$ system delivers competitive $C_2H_4$ electroproduction performance metrics compared to prior reports on acidic C2 electrosynthesis systems (Fig. 3f, Supplementary Table 2)[7,9,11–13,20,22–25], highlighting its potential for scalable acidic $CO_2$ electroreduction.

## COF's role as ion transport regulator

Based on the good performance, we hypothesized that Th-TF COF has two functions: increasing the concentration of $K^+$ and facilitating $OH^-$ migration through its porous channels. To quantify the local $K^+$ concentration at the interface of the Th-TF COF functional layer and catalyst during acidic $CO_2RR$, we employed in-situ operando X-ray fluorescence spectroscopy (XRFS). Supplementary Fig. 19 displays the XRF spectrum of Ar K, K Kα, and Kβ peaks across a $K^+$ concentration gradient (0.5–3 M). A linear baseline correlating the $K^+$/Ar peak ratio to bulk $K^+$ concentration enabled the concentration of $K^+$ within the reactive catalyst area to be calculated. Dynamic in-situ XRFS measurements (Fig. 4a) revealed the evolution of solvated $K^+$ content on

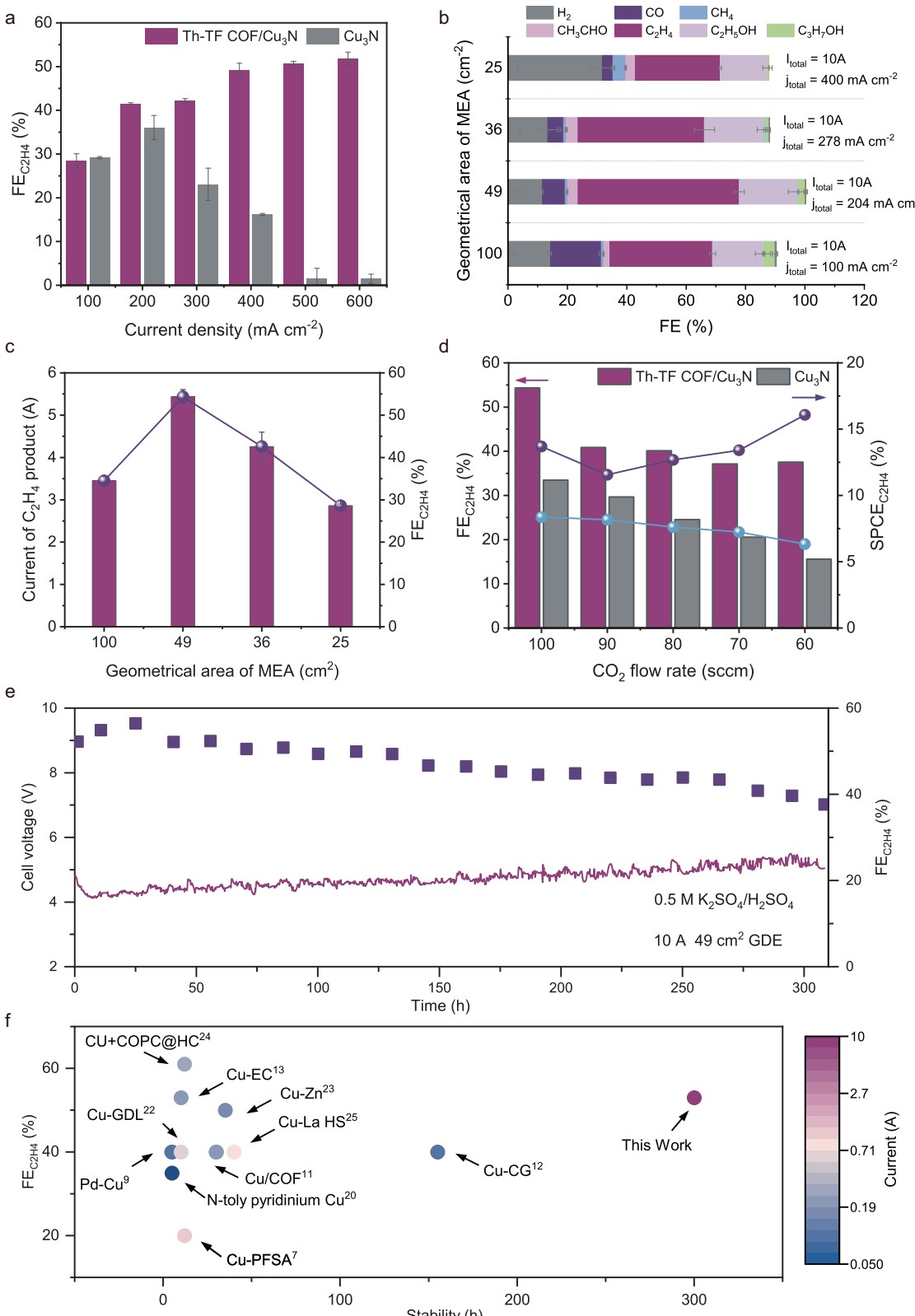

**Fig. 3 | Performance of C₂H₄ electrosynthesis in the acid-fed Th-TF COF AEM MEA system.** **a** C₂H₄ selectivity on CNCP and Sus/Cu₃N electrodes tested in various current densities in H₂SO₄ solution with 0.5 M K₂SO₄ (pH - 1). Values are means, and error bars indicate s.d. (*n* = 3 replicates). **b** CO₂R product distribution and (**c**) C₂H₄ production current of CNCP electrodes at 10 A in a scale-up acid-fed MEA system. **d** FE and single-pass carbon efficiency of CO₂-to-C₂H₄ on 49 cm⁻² CNCP and Sus/Cu₃N electrodes under total current of 10 A (current density = 204 mA cm⁻²) with

different flow rate of CO₂. **e** The system stability performance of CO₂R to C₂H₄ on CNCP GDEs in a scale-up MEA system at a constant current of 10 A. **f** Comparison of this work with previous studies on the acidic electrocatalytic CO₂ to multi-carbon products at a similar electrolyte pH. No iR correction was applied. Source data for Fig. 3 are provided as a Source Data file.

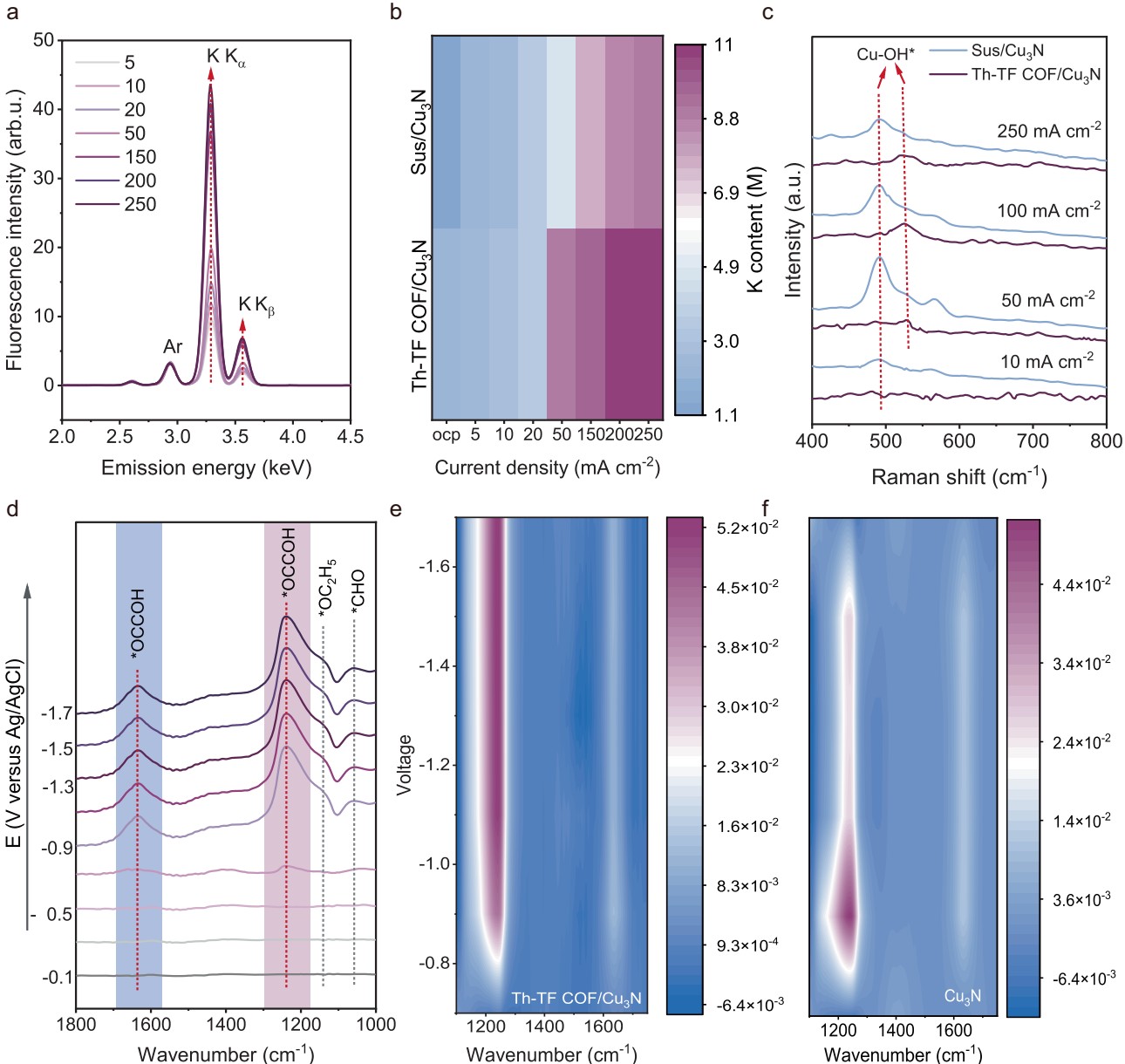

**Fig. 4 | Spectroscopic evidence of COF's role as ion transport regulator. a** XRF spectrum on Th-TF COF/Cu₃N electrode showing the solvated K emissions at different current densities. **b** The local content of K of Th-TF COF/Cu₃N and bare Cu₃N electrodes. **c** In-situ Raman spectra during electrosynthesis of C₂H₄ over Th-TF COF and Sustainion-coated Cu₃N catalysts at different current densities. Electrolyte is 0.5 M K₂SO₄/H₂SO₄ (pH-3). **d** In-situ ATR-SEIRS spectra recorded during acidic CO₂R on Th-TF COF/Cu₃N catalyst from −0.3 to −1.7 V versus Ag/AgCl. Contour map of in-situ ATR-Fourier transform infrared spectra on the Th-TF COF/Cu₃N (**e**) and bare Cu₃N catalysts (**f**) recorded from 1100 to 1750 cm⁻¹. No iR correction was applied. Source data for Fig. 4 are provided as a Source Data file.

the Th-TF COF/Cu₃N electrode under varying current densities. Th-TF COF exhibits ~2.4× higher K⁺ content versus Sustainion at open-circuit potential (OCP, Fig. 4b), suggesting that its cation-adsorption sites promote K⁺ diffusion from the AEM to the catalyst. Under an applied electric field, Th-TF COF/Cu₃N shows stronger K⁺ enrichment than Sus/Cu₃N, particularly at elevated current densities, confirming its role in stabilizing near-surface K⁺ during CO₂RR.

In-situ Raman spectroscopy was used to track the presence of OH⁻ in Th-TF COF. As shown in Fig. 4c, the band located at ~530 cm⁻¹ is associated with the adsorbed OH⁻ on Cu-based catalysts[26], while the red shift of the Cu−OH band on Sus/Cu₃N is due to the interaction with the polymer. Comparing the OH⁻ signal in the Th-TF COF/Cu₃N electrode versus the Sus/Cu₃N electrode as a function of current density in a 0.5 M K₂SO₄/H₂SO₄ (pH-3), it can be seen that only a weak OH⁻ adsorption peak is identified in the Th-TF COF/Cu₃N electrode. This

suggests that the Th-TF COF layer promotes the migration of locally generated OH⁻ outward and thereby maintains a low OH⁻ local environment near the catalyst surface. Another evidence for the low local pH can be judged from the adsorption configuration of CO on the Th-TF COF/Cu₃N electrode. Previous studies revealed that CO is adsorbed in the atop configuration (CO$_{atop}$) on Cu sites in a low pH environment, and converts to the bridge configuration (CO$_{bridge}$) as the pH increases[27]. Supplementary Fig. 20 shows that only the CO$_{atop}$ at 2044-2076 cm⁻¹ is visible between 50 mA cm⁻² to 250 mA cm⁻² on the Th-TF COF/Cu₃N electrode, evidencing the low pH local environment in the COF layer. In contrast, at a higher pH environment, the CO$_{bridge}$ (at 1819–1830 cm⁻¹) is dominant, which was observed for the ionomer-coated electrode. In-situ Raman spectra in the 900–1200 cm⁻¹ region reveal that HCO₃⁻ (~1001–1006 cm⁻¹) and CO₃²⁻ (~1063–1069 cm⁻¹) bands increase in intensity with increasing current density on Sus/

Cu₃N, whereas these signals are largely absent on Th-TF COF/Cu₃N, which indicate suppressed carbonate accumulation and a less alkaline interfacial environment (Supplementary Fig. 21).

In-situ attenuated total reflectance-Fourier transform infrared (ATR-TIR) spectroscopy was employed to investigate the intermediates formed during the $C_2H_4$ electrosynthesis process. Beyond intermediate species, the spectra also provide insight into CO adsorption. Supplementary Fig. 22 shows that the Th-TF COF/Cu₃N electrode exhibits a pronounced $*CO_{atop}$ band (2040–2070 cm⁻¹), whereas bare Cu₃N predominantly shows $*CO_{bridge}$ adsorption (1815–1830 cm⁻¹), consistent with CO configurations on Cu catalysts under acidic conditions. Regarding C-C coupling intermediates, *OCCOH emerges as a key species for $C_2H_4$ formation. As depicted in Fig. 4d, e, a characteristic peak corresponding to *OCCOH appears at 1238 and 1631 cm⁻¹ on Th-TF COF/Cu₃N catalyst[17]. The intensity of the *OCCOH peak increases with the application of voltage. In contrast, the peak of *OCCOH intensity appears weak on the bare Cu₃N catalyst at −1.1 V and gradually diminishes by -1.7 V. The peak area of *OCCOH ($A_{OCCOH, TH-TF COF/Cu3N}/A_{OCCOH, Cu3N}$) in Th-TF COF/Cu₃N is -1.3 times greater compared to Cu₃N from −1.1 V to −1.7 V (Supplementary Fig. 23). This indicates the Th-TF COF on the Cu₃N catalyst allows *OCCOH to be stabilized and facilitates $C_2H_4$ production. Besides, the peaks of *CHO associated at 1057 cm⁻¹ and $*OC_2H_5$ identified at 1130 cm⁻¹ increase with potential, which are recognized as intermediates of $C_2H_5OH$[28,29], further confirming enhanced C₂ selectivity on Th-TF COF/Cu₃N catalyst.

## Molecular dynamics (MD) modeling

To gain insights on the regulation capabilities of Th-TF COF for ions (K⁺, OH⁻) and $CO_2$ molecules, MD simulations were performed (details in Supporting Information). Three models were compared: the bare Cu catalyst (Model I), Cu coated with a Sustainion ionomer (Sus/Cu, Model II), and Cu coated with Th-TF COF (Th-TF COF/Cu, Model III), as illustrated in Supplementary Fig. 24 and described in Supplementary Data 1–4.

## Ion and CO₂ microenvironment

Under concentration gradients and electric fields, K⁺ migrated toward the catalyst surface while OH⁻ diffused into the bulk solution. The local [K⁺]/[OH⁻] ratio, derived from ion density calculations, reflects the catalytic microenvironment influencing the $C_2H_4$ conversion pathway in acidic $CO_2RR$. Simulations (Fig. 5a, Supplementary Figs. 25, 26) revealed a low [K⁺]/[OH⁻] ratio of 0.46 near the bare Cu surface (<1 nm). Introducing the Sustainion ionomer (Model II) enhanced outward OH⁻ transport via anion-exchange nanochannels, quadrupling the [K⁺]/[OH⁻] ratio relative to bare Cu. Replacing ionomer with Th-TF COF (Model III) further increased the ratio to 3.2, highlighting its superior ability to promote OH⁻ migration and restrict K⁺ flux. Local $CO_2$ concentrations ($[CO_2]$) also differed significantly across models. As shown in Fig. 5b, Supplementary Fig. 27, Th-TF COF/Cu maintained higher $[CO_2]$ near the catalyst compared to bare Cu and Sus/Cu. This may arise from $CO_2$ adsorption by the hydrazone linkages (C = N − NH) in the Th-TF COF backbone, enriching $CO_2$ at the catalyst interface.

## Diffusion dynamics via mean square displacement (MSD)

MSD calculations (Fig. 5c, Supplementary Fig. 28) demonstrate that Th-TF COF/Cu accelerates OH⁻ diffusion ($1.5 \times 10^{-5}$ cm² s⁻¹) to the anion-exchange membrane (AEM), yielding a diffusion coefficient -2.5× higher than bare Cu ($0.6 \times 10^{-5}$ cm² s⁻¹) and Sus/Cu ($0.7 \times 10^{-5}$ cm² s⁻¹). Conversely, K⁺ diffusion in Th-TF COF/Cu was slower ($0.59 \times 10^{-5}$ cm² s⁻¹) than in bare Cu ($0.83 \times 10^{-5}$ cm² s⁻¹) and Sus/Cu ($0.78 \times 10^{-5}$ cm² s⁻¹), confirming K⁺ confinement and the formation of a K⁺-rich microenvironment. Critically, $CO_2$ transport from the gas diffusion layer (GDL) to the catalyst/electrolyte interface was most efficient in Th-TF COF/Cu, with a diffusion coefficient ($D_{CO_2}$) of

$1.9 \times 10^{-5}$ cm² s⁻¹, surpassing values for bare Cu ($D_{CO_2} = 1.3 \times 10^{-5}$ cm² s⁻¹) and Sus/Cu ($D_{CO_2} = 1.4 \times 10^{-5}$ cm² s⁻¹).

MD simulations demonstrate that Th-TF COF regulates K⁺/OH⁻ fluxes by selectively restricting K⁺ diffusion while enhancing OH⁻ transport and $CO_2$ mass transfer. These findings align with in situ characterization data, corroborating Th-TF COF's dual role in ion/molecule regulation for acidic $CO_2RR$.

## Ion transport mechanisms in Th-TF COF channels

To further investigate ion interactions with the Th-TF COF framework, we simulated dynamic bidirectional flow through its nanochannels. As shown in Fig. 5d, under acidic conditions, the C = N − NH and C−N groups on the pristine Th-TF COF skeleton undergo protonation, forming positively charged sites ($-NH_2^+$), hereafter termed "protonated Th-TF COF." We hypothesized that these charged groups facilitate selective ion migration. To test this, we modeled anion/cation transport pathways through pristine and protonated Th-TF COF (models showed in Supplementary Fig. 29) under electric field and concentration gradients.

## OH⁻ Migration Dynamics

In protonated Th-TF COF (Fig. 5e, f), axial OH⁻ migration is favored due to (i) shorter hopping distances and (ii) wider transport channels enhanced by adjacently aligned H atoms from protonated sites along the axial pathway. Electrogenerated OH⁻ at the catalyst surface migrates through COF channels via hydrogen-bond interactions with protonated groups (Eq. 2), either reacting with H⁺ at the AEM interface to form $H_2O$ (Eq. 3) or diffusing toward the anode. The MSD results confirm that protonated functional groups significantly enhance OH⁻ mobility: protonated Th-TF COF/Cu exhibits a diffusion coefficient of $1.5 \times 10^{-5}$ cm² s⁻¹, 1.8× higher than the non-protonated model ($0.8 \times 10^{-5}$ cm² s⁻¹) (Supplementary Figs. 30, 31). Driven by electric field, the electrogenerated OH⁻ near the catalyst surface migrates through the COF channels via H−bond interaction, and partially encounters with the H⁺ and forming the $H_2O$ at the AEM interface or traverses to the anode side (Eqs. (1)−(3)).

$$Cathode: 2CO_2 + 8H_2O + 12e^- \rightarrow C_2H_4 + 12OH^- \quad (1)$$

$$COF\ channel: OH^- + C = N - NH_2^+ \rightleftharpoons H - O \cdots H_2N - N = C \quad (2)$$

$$AEM\ interface: 12OH^- + 12H^+ \rightarrow 12H_2O \quad (3)$$

In contrast, K⁺ migration is governed by interactions with keto groups (C = O) on the COF backbone. These groups act as preferential binding sites for K⁺, promoting ion enrichment near the catalyst surface while restricting bulk diffusion (Supplementary Fig. 32). This selective retention creates a K⁺-rich microenvironment, critical for stabilizing reaction intermediates in acidic $CO_2RR$.

## Discussion

We have demonstrated the use of Th-TF COF as a cathode modifier in an acid-fed AEM MEA system to enhance $CO_2$-to-$C_2H_4$ and energy conversion efficiency. Th-TF COF serves as a multifaceted ion transport regulator in acidic $CO_2RR$, optimizing the catalytic microenvironment through dual functionalities: (1) enriching K⁺ near the catalyst surface and (2) accelerating OH⁻ migration away from the reaction interface. Experimental and computational evidence demonstrates that its porous framework, functionalized with cation-adsorbing keto groups and protonated hydrazone linkages, selectively confines K⁺ while facilitating OH⁻ diffusion via hydrogen-bond-mediated pathways. Concurrently, the COF enhances $CO_2$ mass transfer through adsorption, enriching reactant availability. These synergistic effects promote the stabilization of critical intermediates

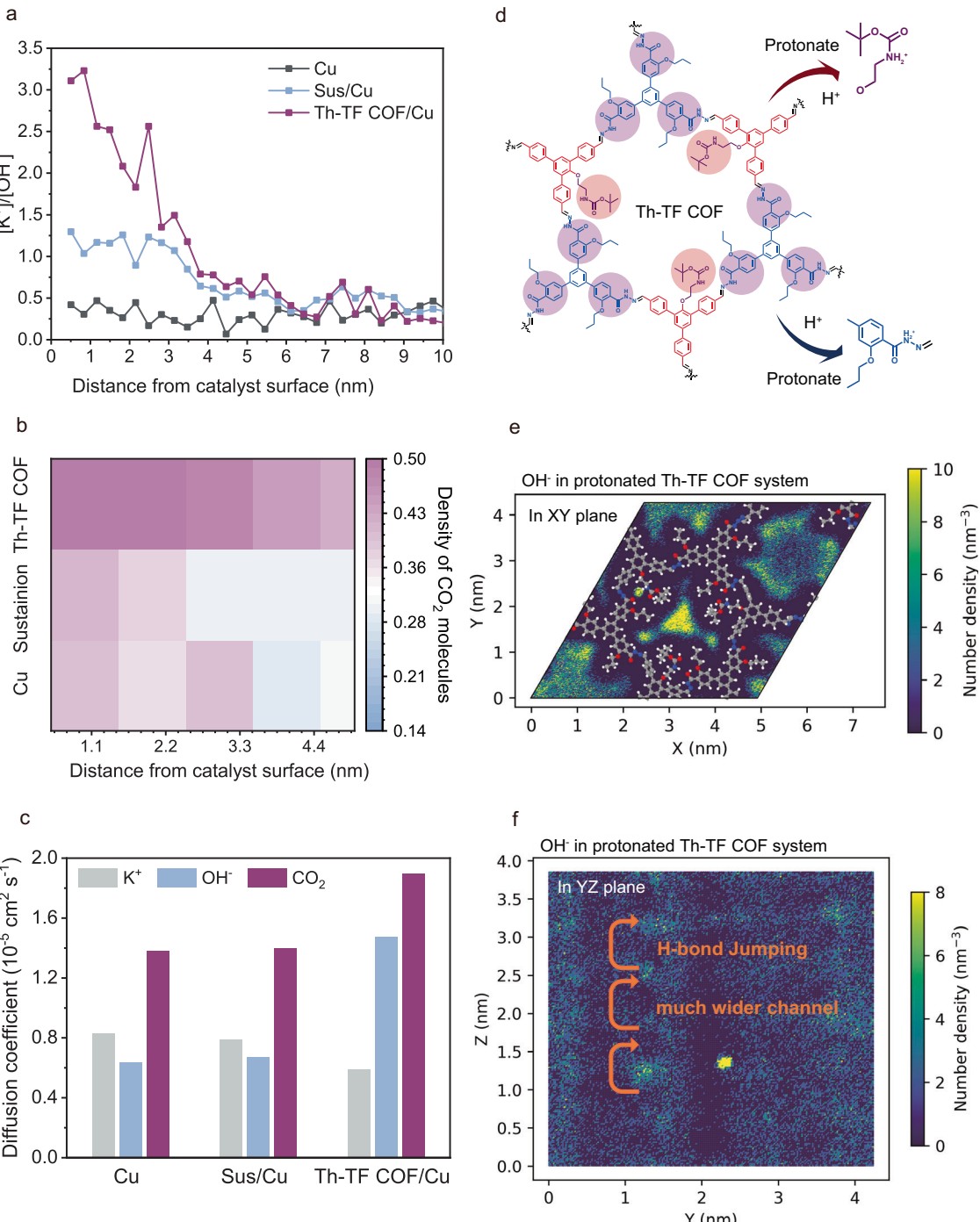

**Fig. 5 | Molecular dynamics (MD) simulations. a** Ratio of $K^+$ to $OH^-$ concentration and (**b**) distribution of $CO_2$ molecules at different distances along the catalyst surface under an applied electric field for Cu, Sus/Cu, and Th-TF COF/Cu models. **c** Self-diffusion coefficients of $K^+$, $OH^-$, and $CO_2$ derived from mean square displacement (MSD) analysis. **d** Structure of Th-TF COF and its protonation process under acidic conditions. Two-dimensional density contour maps of protonated Th-TF COF under simulated acidic $CO_2RR$ conditions, revealing reactive sites (**e**) and ionic migration pathways (**f**) within the channels. Detailed MD simulation parameters and procedures are described in the Methods and Supporting Information. Source data for Fig. 5a–c and Fig. 5e, f are provided as a Source Data file.

(e.g., *OCCOH) and favor $C_2$ product formation. Our studies show that careful design of functionalities in porous materials, such as COF, allows cation and anion transport dynamics to be decoupled, offering a strategy for ion-regulating frameworks in electrocatalytic systems.

## Methods

### Synthesis of Cu₃N catalyst

The synthesis method of $Cu_3N$ has been reported previously[17]. Notably, 4 g of copper acetate ($Cu(COOH)_2$) powder was ground by ball milling (BM) for 2 h at the speed of 400 reps before synthesis. In a typical synthesis of $Cu_3N$, 200 mg of BM-$Cu(COOH)_2$ and 2 g of urea were placed in two different porcelain boats and put into a tube furnace with urea at the upstream side. Then, the furnace was heated to 250 °C at a heating rate of 2 °C/min and maintained at 250 °C for 1 h with Ar gas flowing at 30 sccm. The resulting dark brown powder product was washed with ethanol/water and dried overnight at 60 °C to get $Cu_3N$. All chemicals were purchased from Sigma-Aldrich and used as received unless otherwise specified.

## Synthesis of Th-TF COF

**Synthesis of ThzOPr.** The synthesis method of ThzOPr has been reported previously[30]. To a mixture of methyl 2-hydroxy-4-iodobenzoate (2.78 g, 10 mmol), $K_2CO_3$ (5.6 g, 40 mmol), and KI (100 mg, 0.6 mmol) in acetone (150 mL) was added with 1-propylbromide (2 mL, 22 mmol) dropwisely. The mixture was refluxed with stirring under $N_2$ atmosphere for 2 days and hot filtered through a Celite bed. The filtrate was evaporated and purified via flash chromatography (hexane/ethyl acetate = 5:1) to give a colorless oil. The oil was then added to a mixture of bis(pinacolato)diboron (2.67 g, 10.5 mmol), potassium acetate (2.94 g, 30 mmol), $PdCl_2(PPh_3)_2$ (105 mg, 0.15 mmol) in 1,4-dioxane (50 mL) under $N_2$ atmosphere. The mixture was heated at 100 °C for 18 h. The mixture was diluted with ethyl acetate and water. The organic layer was separated, dried over $Na_2SO_4$ and evaporated via vacuum. The residue was purified by flash chromatography (hexane/ethyl=5:1) to give a colorless oil (2.59 g, 81%). Then to a mixture of 1,3,5-tribromobenzene (315 mg, 1 mmol), $K_2CO_3$ (912 mg, 6.6 mmol), and $Pd(PPh_3)_4$ (116 mg, 0.1 mmol) in 1,4-dioxane/$H_2O$ (18 mL, 5:1), methyl 2-propoxy-4-(4,4,5,5-tetramethyl-1,3,2-dioxaborolan-2-yl)benzoate (1 g, 3.3 mmol) was added under $N_2$. The mixture was refluxed for 2 days. After the reaction was cooled to room temperature, the mixture was added with ethyl acetate and water. The organic layer was separated and dried over $Na_2SO_4$. The solvent was removed by vacuum, and the residue was purified by flash chromatography (hexane:EA = 10:1 then $CH_2Cl_2$) to give a light brown solid. The solid was suspended in ethanol (10 mL) and added with hydrazine monohydrate (1.5 mL). The mixture was refluxed for 1 day. The precipitate was filtered, washed with ethanol and dried to give a white solid (445 mg, 68%).

**Synthesis of TFPBr.** To a mixture of ethanolamine (726 μL, 12 mmol) in 28 mL dry DCM was added with NEt3 (2.51 mL). The mixture was cooled down to 0 °C and di-tert-butyl decarbonate (2.62 g, 12 mmol) in DCM (7 mL) was added dropwise and stirred at room temperature overnight, the product denoted as **A**. To a mixture of 2,4,6-Tribromophenol (992 mg, 3 mmol), Triphenylphosphine (1.53 g, 6 mmol), and 1,1'-(Azodicarbonyl)dipiperidine (1.514 g, 6 mmol) was added into dry THF (8 mL). The mixture was cooled down to 0 °C, and a solution of **A** (580 mg) was added dropwise to dry THF (2 mL). The mixture was then warmed to room temperature and heated at 60 °C for 2 days. The filtrate was evaporated and purified via flash chromatography (hexane/ethyl acetate = 50:1) to give a colorless oil. The oil (237 mg) was then added to a mixture of 4-Formylphenylboronic acid (247 mg, 1.65 mmol), Tetrakis(triphenylphosphine) palladium (0) (17.3 mg, 0.015 mmol), and $K_2CO_3$ (829 mg, 6 mmol) into THF (6 mL). The mixture was degassed and heated at 80 °C under an Ar atmosphere. The residue was purified by flash chromatography (hexane/ethyl = 10:1) to give a colorless oil.

**Synthesis of the Th-TF COF.** To a 10 mL Schlenk tube (15 mm × 80 mm) was added with 1,3,5-triformylbenzene (0.45 ml) and 1,4-Dioxane (0.05 ml). The mixture was sonicated for 5 mins, added with 6 M acetic acid (100 μL), flash frozen at 77 K, and degassed under freeze-pump-thaw for three cycles. The tube was then sealed and heated at 120 °C in an oven for three days. The solid obtained was exchanged with THF (5 mL) for 5 times and dried under vacuum to afford corresponding COF.

## Charaterization

Atomic-resolution annular dark-field scanning transmission electron microscopy (ADF-STEM) was performed using an aberration-corrected JEOL JEM-ARM200F transmission electron microscope, operated at 200 kV. Energy-dispersive X-ray spectroscopy (EDS) mapping was carried out on a Thermo Fisher Spectra 300 microscope, also operated at 200 kV. The X-ray diffraction (XRD) patterns were collected on a Bruker D-8 instrument (Cu Kα radiation, λ = 0.154056 nm) at room temperature. X-ray photoelectron spectroscopy (XPS) was collected using Al Kα radiation, hν = 1486.6 eV, on a Thermo Fisher Scientific ESCALAB Xi+ instrument.

## Electrode preparation

10 mL isopropanol was used as a solvent to disperse both $Cu_3N$ catalyst ($\sim$1.0 mg cm$^{-2}$) and Sustainion (100 uL). COF-modified $Cu_3N$ electrodes (CNCP) were prepared by spray-coating COF nanoparticles onto the pristine $Cu_3N$ gas diffusion electrodes (GDEs). To facilitate uniform spray coating, COF nanoparticles were dispersed in 15 mL isopropanol and sonicated for at least 1 h. Unless otherwise stated, the nominal loading of COF nanoparticles on the carbon paper was set at approximately 0.5 mg cm$^{-2}$. Finally, 200 μL of Sustainion XA-9 ionomer dissolved in ethanol (5 wt%) was spray-coated onto the electrode surface. The effect of interfacial K concentration on acidic $CO_2RR$ was studied by coating $Cu_3N$ catalyst with different concentrations of potassium ions. 10 mL aqueous solution containing different molar concentrations (0–0.3 M) of $K_2SO_4$ were spray-coated onto $Cu_3N$ GDEs.

## MEA-cell assembly

For MEA electrolyzers, cathode and anode flow fields were used with the active areas of 1, 25, 36 and 49 cm$^2$, corresponding to the $1 \times 1$, $5 \times 5$, $6 \times 6$ and $7 \times 7$ cm$^2$ electrode windows, respectively. IrOx-Ti mesh was used as the anode, an anion exchange membrane (Pipeion), and COF-modified $Cu_3N$ GDE as the cathode electrode. The cell was operated under ambient temperature and pressure. Dry $CO_2(g)$ was supplied from the back side of the cathode, and 0.5 M $K_2SO_4$/$H_2SO_4$ was circulated in the anode part. The scale-up MEA cell stability test in an acidic system is assembled by using $7 \times 7$ cm$^2$ electrode windows at a constant current of 10 A. The total cathode area was 49 cm$^2$, and the flow rate of the $CO_2$ inlet was 100 sccm.

## Electrochemical measurement

All electrochemical tests were performed using an electrochemical workstation (Autolab PGSTAT302N) connected to a current booster (Metrohm Autolab,10 A). The catholyte of pH ~1.0, 2.0, and 4.0 were prepared by introducing a 0.5 M $K_2SO_4$ into specific amount of sulfuric acid was used as anolyte. For neutral electrolytes preparation (pH ~ 6.3–6.8), a 0.5 M $K_2SO_4$ solution was used directly. The $CO_2RR$ performance was tested in MEA-cell assemblies under galvanostatic mode. The current densities reported are based on the geometric surface areas. No iR correction was applied to the electrochemical data presented in this study.

## CO₂R product analysis

The gas products were collected from the gas outlet of the MEA cell, which were injected into a gas chromatograph for gas quantification. The gas chromatograph was equipped with a thermal conductivity detector for the detection of $H_2$ and CO signals and a flame ionization detector for the detection of $CH_4$ and $C_2H_4$ signals. The gas chromatograph was composed of packed columns of molecular sieves (5 Å) and Carboxen-1000 and employed Argon (99.999%) as the carrier gas. and liquid products from $CO_2RR$ were measured by high-performance liquid chromatography (YL9100) and headspace GC (YL Instruments). The FE was calculated using the equations:

$$FE_{gas} = \frac{z \times F \times v \times r}{j \times V_m} \text{ and } FE_{liquid} = \frac{z \times F \times n_{product}}{Q}$$

where z is the number of electrons transferred, F is the Faraday's constant (96,485 C mol$^{-1}$), v is the gas flow rate at the outlet of the gas chamber (l min$^{-1}$), r is the concentration of detected gas product in parts per million, j is the total current (A), $V_m$ is the unit molar volume

of gas (24.5 L mol$^{-1}$), nproduct is the total moles of product derived from headspace GC analysis, and Q is the total charge (C).

The $CO_2$ SPCE towards each product was determined using the following equation at 25 °C, 1 atm:

$$SPCE = \frac{(j_{product} \times 60s)/(n \times F)}{(v \times 1\,min)/V_m}$$

where $j_{product}$ is the partial current (A) of a specific $CO_2RR$ product, n is the electron transfer for the formation of each product molecule, and $V_m = 24.5\,L\,mol^{-1}$.

The full-cell energy efficiency for each product was calculated as follows:

$$EE_{product} = \frac{(1.23 + (-E^0_{product})) \times FE_{product}}{-E_{cell}}$$

Where $E^0_{product}$ is the thermodynamic potential for the formation of a specific $CO_2RR$ product, FEproduct is the calculated FE of the product and $E_{cell}$ is the full-cell voltage without Ohmic loss correction evaluated in the MEA cells.

## Data availability

All data are available from the authors upon reasonable request. Source data are provided with this paper.

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

## Acknowledgments

K.P.L. acknowledges funding support from Singapore-International Synchrotron Access Program (SG-ISAP) as well as funding support from Center for Hydrogen Innovations CHI-P2022-01 and Singapore's Ministry of Education Tier 1 Grant A8002669-00-00. Q.H.Y. acknowledges funding support from the National Natural Science Foundation of China (Nos. 52432005).

## Author contributions

D.C. conceived the research, synthesized the materials, and conducted catalytic measurements under the supervision of K.P.L. and Q.H.Y. The

COF monomer designed were conducted by Y.Y.; STEM-HAADF and HRTEM were performed by X.H.; SEM was performed by Z.K.; FTIR measurements were performed by S.H.; Large current flow cell operation was performed with the assistance from J.L. and Q.H.; XAFS was performed by S.X.; The draft was written by D.C., revised by K.P.L. All authors discussed and commented on the manuscript.

## Competing interests

The authors declare no competing interests.
