## [Transparent Peer Review file · Nature Communications]

Electrocatalytic CO₂ Reduction to Ethylene in An Acid-fed Membrane Electrode Assembly at 10A

Corresponding Author: Professor Kian Ping Loh

Version 0:

Reviewer comments:

Reviewer #1

(Remarks to the Author)

This manuscript entitled "Electrocatalytic CO₂ Reduction to Ethylene in An Acid-fed Membrane Electrode Assembly at 10A" presents an acid-fed MEA system designed for highly efficient ethylene electrosynthesis, demonstrating high Faradaic efficiency and energy efficiencies for C₂H₄. The 100 cm² COF-functionalized MEA electrolyzer operates steadily at 10 A with stability surpassing 300 hours. In situ analyses combined with molecular dynamics simulations elucidate the reaction pathways and ion-regulation mechanism of COF materials. However, revisions such as additional control experiments and structural characterizations are suggested to strengthen the novelty of manuscript and mechanistic rigour. Therefore, publication is recommended pending these revisions.

1. The authors propose employing an AEM (Fig. 1) in the acid-fed MEA system to facilitate rapid OH⁻ ion transport and enhance stability by mitigating salt precipitation. However, detailed experimental analysis of the anode tail gas is needed is required to confirm the reduction or elimination of unreacted CO₂ crossover compared to neutral MEA systems.
2. Dose Cu₃N undergo structural reconstruction during CO₂ reduction? Comprehensive in situ or post-reaction analyses (e.g., XPS, XRD, TEM) are required to determine the degree of catalyst reconstruction or transformation.
3. The authors detail the screening and design approach for the Th-TF COF (Figs. S8 and S9). Further characterization of this COF is recommended to verify the presence of specific functional groups.
4. Given that COF material also possesses hydrophobic properties, have the authors compared them with conventional hydrophobic polymers to distinguish between its ion-regulating abilities and hydrophobic characteristics?
5. If the Th-TF COF acts as a cathode modifier in an acidic MEA system, how does it perform in acidic flow cell conditions?
6. Although the authors achieved long-term cycling stability with COF, only SEM images of the electrode were provided (Fig. S15). Please include FTIR spectra of the COF before and after the reaction to clear illustrate its chemical stability.

Reviewer #2

(Remarks to the Author)

In the present manuscript, the author elucidates the attainment of efficient and stable electrocatalytic reduction of CO₂ to ethylene through the integration of a thiolate-terminated Th-TF covalent organic framework (COF) with copper nitride (Cu₃N) catalysts within an acidic feed membrane electrode assembly (MEA) system. This approach markedly augments the selectivity towards multi-carbon products and enhances the energy conversion efficiency of the system. While the strategy exhibits innovative potential, several pivotal mechanistic issues necessitate further clarification in its current implementation. If the author can successfully address the following issues, this manuscript can be reconsidered.

1. The author's choice of copper nitride (Cu₃N) as the catalyst and Th-TF COF as the ion transport regulator, culminating in exceptional performance in electrocatalytic CO₂ reduction to ethylene, is indeed innovative. Nevertheless, supplementary characterization data concerning the morphology, structure, and valence state of the Th-TF COF/Cu₃N composite post-reaction are imperative to substantiate its stability throughout the reaction, ensuring no restructuring or other alterations occur.
2. The author posits that Th-TF COF mitigates local pH elevation by expediting OH⁻ migration but falls short of quantifying the impact of ionic transport resistance on system performance. Under acidic conditions, the efficacy of OH⁻ migration from the cathode to the anode may be constrained by membrane resistance and the COF-channel architecture. The above issues need to be clarified
3. EIS data should be supplemented to elucidate the relationship between OH⁻ migration rate and membrane resistance,

analyzing the charge transfer resistance (R_{ct}) and membrane resistance (R_m) of the MEA at varying current densities. This will verify whether Th-TF COF accelerates OH⁻ migration by diminishing R_m .

4. Molecular dynamics simulations ought to be employed to elucidate how the COF-channel aperture (e.g., lattice spacing of approximately 3.5 Å as depicted in MD simulations) selectively permits OH⁻ passage while impeding K⁺ diffusion (refer to Figure 5). The actual channel dimensions should be corroborated via pore size distribution measurements (e.g., N₂ adsorption-desorption experiments).

5. In Figure 4c, the author utilizes in-situ Raman spectroscopy to monitor OH⁻ presence within the Th-TF COF. By comparing the intensity variations of adsorbed OH⁻ at 530 cm⁻¹, the author demonstrates enhanced OH⁻ adsorption on Th-TF COF/Cu₃N relative to Sus/Cu₃N. However, the superior catalytic performance of Th-TF COF/Cu₃N necessitates a comprehensive explanation for this phenomenon.

6. The in-situ Raman spectrum presented in Figure 4c is confined to the 400-800 cm⁻¹ range. Nonetheless, the quantity of surface-adsorbed OH⁻ significantly influences the local pH near the electrode, which can be further elucidated by comparing the HCO₃⁻ signal at 1011 cm⁻¹ and the CO₃²⁻ intensity ratio at 1067 cm⁻¹ in the Raman spectrum. The author is thus requested to furnish this segment of the Raman spectrum for a more holistic proof.

7. In Supplementary Figure 17, the authors delineate the lower local pH environment within the COF layer by comparing the variation trends of *CO_{atop} and *CO_{bridge} under diverse current densities. However, the in-situ infrared spectra depicted in Figure 4d and Supplementary Figure 18 lack spectral data at this wavenumber. The authors are, therefore, requested to provide the infrared spectra for this section to more robustly substantiate their findings.

8. In Figure 4d, the authors demonstrate that the Th-TF COF stabilizes *OCCOH and fosters ethylene formation on the Cu₃N catalyst by comparing the peak area ratios of *OCCOH. Nevertheless, in Supplementary Figure 18, the *OCCOH signal at 1238 cm⁻¹ is notably higher for Cu₃N than for Th-TF COF/Cu₃N at potentials ranging from -0.3 V to -0.7 V. Does this imply that the conclusion regarding the Th-TF COF's ability to stabilize *OCCOH and promote ethylene formation is invalid under low overpotential conditions? An explanation for this phenomenon is warranted.

9. The author asserts that the Th-TF COF on the Cu₃N catalyst stabilizes *OCCOH and promotes ethylene formation. However, the in-situ infrared spectrum in Figure 4d detects an ethanol intermediate. Given that ethylene and ethanol formation constitute a competitive process, the author is requested to elucidate why Th-TF COF/Cu₃N selectively promotes ethylene formation over ethanol. Additional experiments or theoretical calculations are requisite to substantiate this claim.

Reviewer #3

(Remarks to the Author)

This work aimed to solve the problems of low ethylene selectivity and stability in CO₂ reduction systems operating in acidic conditions. They designed a AEM system used a COF layer on the cathode to reduce local OH⁻ concentration and enrich K⁺ near the catalyst. Comments:

1. The novelty of this work is not clear. Similar strategies using COFs to create a proton-blocking and K⁺-rich environment for improving C₂⁺ selectivity have been reported. That approach is established.

2. I appreciate the MEA approach, but I don't think it is workable in acidic with a membrane. That is why flow cells and slim flow cells have been the norm in the field of acidic CO₂R.

3. Related to the above is the concern regarding CO₂ crossover. (Bi)carbonate species may still form and migrate through the anion exchange membrane during operation, even under acidic conditions. Quantify and report single pass CO₂ conversion (and thus also crossover loss).

4. The CNCP system shows a quite high cell voltage (4.2–5.0 V at 200 mA cm⁻²) and only moderate C₂H₄ selectivity (~50%). These values appear less competitive when compared to the best flow-cell configurations that are between 3-4V with higher FE.

5. The long-term stability claims may be overstated: "... a full-cell voltage of ~4.5 V under a total current of 10 A (current density of 204 mA cm⁻²) achieved a Faradaic efficiency of 50% for CO₂-to-C₂H₄ conversion with long-term stability over 300 h." Yet in Fig. 3e, FE drops below 50% well before 100 h, and the voltage increases to 5 V around 100 h.

6. The reasons for the improved stability are not clearly explained, and the explanation for the performance decay isn't convincing. According to Fig. 3e, both FE and voltage decline steadily at a similar rate over the 300 h test, which doesn't quite match the typical behavior of GDL flooding.

7. There are errors that affect readability: In Figs. 2d and 3b, the yellow portion of the product distribution is undefined. Reference labels in Fig. 3f are wrong (pointing to the wrong references). In Fig. 4a, the unit for potassium concentration is missing.

Thus the work does not meet the high standards of Nat Communications, in my view. I hope these comments are useful for the authors in submission elsewhere.

Version 1:

Reviewer comments:

Reviewer #1

(Remarks to the Author)

The author has satisfactorily addressed all my questions. I have no further comments and recommend that this paper be published.

Reviewer #2

(Remarks to the Author)

The authors have addressed the comments and the manuscript is now acceptable.

Reviewer #3

(Remarks to the Author)

I appreciate the revisions. I continue to worry about the SPC achieved, but the MS is clear on the values. Based on other merits and reviews/responses, I recommend publication.

Manuscript: NCOMMS-25-44692-T

Title: “Electrocatalytic CO₂ Reduction to Ethylene in An Acid-fed Membrane Electrode Assembly at 10A”

The authors greatly appreciate reviewers’ insightful comments and careful review on our manuscript (NCOMMS-25-44692-T). This manuscript has been revised carefully according to the comments of the reviewers. The responses are listed point-by-point in the following contents, and revisions have been highlighted by yellow color in the revised manuscript. Following are our responses and detailed explanation towards these comments from the reviewers.

Responses to Reviewers:

To Reviewer #1:page 2—page18

To Reviewer #2:page19—page44

To Reviewer #3:page45—page69

Response to Reviewers' comments

Reviewer #1:

This manuscript entitled "Electrocatalytic CO₂ Reduction to Ethylene in An Acid-fed Membrane Electrode Assembly at 10A" presents an acid-fed MEA system designed for highly efficient ethylene electrosynthesis, demonstrating high Faradaic efficiency and energy efficiencies for C₂H₄. The 100 cm² COF-functionalized MEA electrolyzer operates steadily at 10 A with stability surpassing 300 hours. In situ analyses combined with molecular dynamics simulations elucidate the reaction pathways and ion-regulation mechanism of COF materials. However, revisions such as additional control experiments and structural characterizations are suggested to strengthen the novelty of manuscript and mechanistic rigour. Therefore, publication is recommended pending these revisions.

Response: We sincerely thank the reviewer for their thoughtful comments and constructive suggestions. In this revised version, we have carefully addressed all points raised and made corresponding improvements throughout the manuscript. Specifically, we have supplemented the missing data, including:

1. Quantitative analysis of the anode outlet gas to confirm the suppression of CO₂ crossover in the acid-fed MEA relative to neutral MEA controls.
2. Comprehensive post-reaction characterizations of the Cu₃N catalyst (XRD, TEM, and XAFS) to evaluate its structural evolution and degree of reconstruction during CO₂RR.
3. Additional FTIR spectra of the Th-TF COF before and after long-term operation to confirm the retention of functional groups and chemical stability.

4. Comparative experiments between Th-TF COF and conventional hydrophobic polymers to distinguish the ion-regulating function of the COF from purely hydrophobic effects.
5. Performance benchmarking of the Th-TF COF-modified Cu₃N cathode under acidic flow-cell conditions to evaluate its versatility beyond the MEA configuration.

These additions aim to provide a more complete and mechanistic understanding of **CO₂ crossover mitigation, catalyst reconstruction, COF stability, and the distinction between ion-regulation and hydrophobicity, as well as device-level performance in both MEA and flow-cell architectures.** We believe these revisions significantly strengthen the manuscript and hope it now meets the criteria for publication in *Nature Communications*.

Question 1: *The authors propose employing an AEM (Fig. 1) in the acid-fed MEA system to facilitate rapid OH⁻ ion transport and enhance stability by mitigating salt precipitation. However, detailed experimental analysis of the anode tail gas is needed is required to confirm the reduction or elimination of unreacted CO₂ crossover compared to neutral MEA systems.*

Response: We thank the reviewer for emphasizing the importance of directly verifying CO₂ crossover. To address this, we performed anode tail-gas analysis under neutral (1 M KHCO₃, pH ~7) and acidic (0.5 M K₂SO₄/H₂SO₄, pH ~1) electrolytes at current densities of 100-400 mA cm⁻². As shown in **Fig. R1a**, in the neutral AEM-MEA, the CO₂ fraction in the anode outlet gas rose significantly with current density (~6.1% to ~23.3%), consistent with previous reports of pronounced carbonate crossover in bicarbonate systems.^[1] To exclude contributions from HCO₃⁻ decomposition in the electrolyte itself, we conducted control tests using 1 M KHCO₃ without CO₂ feed, which showed only trace CO₂ signals (0.767-0.918% across 100-400 mA

cm⁻²). By contrast, in **Fig. R1b**, the acidic AEM-MEA displayed much lower CO₂ fractions (~1.2% to ~12.4%) across the same current range. This suppression arises because the proton-rich acidic diffusion layer rapidly reconverts carbonate species (CO₃²⁻/HCO₃⁻) into CO₂ within the cathode vicinity, preventing their migration across the membrane and thereby minimizing CO₂ loss at the anode. These results provide direct experimental evidence that the acid-fed AEM MEA markedly reduces CO₂ crossover compared with neutral systems, underscoring the carbon-efficiency advantage of our design.

Fig. R1 | CO₂ crossover analysis in different AEM-MEA electrolytes at current densities of 100-400 mA cm⁻². (a) Neutral MEA using 1 M KHCO₃ with AEM. The CO₂ fraction in the anode outlet gas increased from ~6.1% to ~23.3% with rising current density, evidencing severe carbonate crossover. (b) Acidic MEA using 0.5 M K₂SO₄/H₂SO₄ with AEM. The CO₂ fraction remained much lower, ranging from ~1.2% to ~12.4% across the same current range. (c) Direct comparison of CO₂ crossover between neutral (1 M KHCO₃) and acidic (0.5 M K₂SO₄/H₂SO₄) electrolytes, highlighting the suppression of CO₂ crossover under acidic conditions. All values are given as volume percentages (%). Gas samples were collected twice within the first 1 h of electrolysis, with N₂ (5 sccm) supplied as a protective carrier gas.

[Ref. 1] Huang, J. E. et al. CO₂ electrolysis to multicarbon products in strong acid. *Science*, **372**, 1074-1078 (2021).

Question 2: *Dose Cu₃N undergo structural reconstruction during CO₂ reduction? Comprehensive in situ or post-reaction analyses (e.g., XPS, XRD, TEM) are required to determine the degree of catalyst reconstruction or transformation.*

Response: We thank the reviewer for raising this important point. To address it, we performed a series of pre- and post-reaction analyses, including ex-situ XRD, operando XANES, and STEM-EDS, on the Cu₃N electrodes employed in the acid-fed MEA tests.

We conducted XRD to examine structural changes before and after CO₂RR. As shown in **Fig. R2a**, prior to electrolysis the electrode displays characteristic Cu₃N reflections at ~23.3°, 40.9°, and 47.6°, corresponding to the (100), (111), and (200) planes, consistent with the pristine

catalyst (**also see in Revised Supplementary Fig. 4**). After prolonged CO₂RR, all Cu₃N peaks vanish, while a broadened feature appears at ~43-45°, assignable to metallic Cu (111). The signals near ~25° and ~55° originate from the carbon paper substrate. These results clearly indicate a phase transformation from Cu₃N into Cu under prolonged reaction conditions.

As shown in **Fig. R2b**, we carried out operando XANES to probe the electronic structure and local coordination environment. As shown, the Cu K-edge XANES spectrum of the post-reaction Cu₃N electrode closely resembles that of metallic Cu. The edge position ($E_0 \approx 8979$ - 8981 eV) aligns with Cu foil, while the white line at ~8995 eV appears slightly weakened and broadened. The near-edge oscillations at higher energies resemble those of Cu foil but with diminished amplitude, suggesting a reduced coordination number and enhanced structural disorder or nano crystallinity. Collectively, these features point to reconstruction of Cu₃N into a predominantly metallic Cu⁰ phase during operation, with any residual Cu-N coordination expected to be minimal and confined to local surface or defect sites. We also performed STEM-EDS to directly visualize the structural motifs. In **Figs. R2c and d**, pristine atomic-resolution ADF/ABF-STEM reveals the ordered Cu lattice and N sublattice (3.5 Å spacing along [010]), consistent with the Cu₃N phase (**see also in Revised Manuscript Fig. 2b**). In **Fig. R2e and f**, after reaction, STEM mapping shows intense Cu signals but markedly diminished N, confirming nitrogen depletion during the structural transition.

Together, these results demonstrate that Cu₃N undergoes reconstruction into metallic Cu under acidic MEA CO₂RR conditions. This observation is consistent with previous reports showing that Cu₃N inevitably reduces to Cu during CO₂RR, while residual Cu-N interfacial motifs may persist and contribute to C-C coupling activity.^[1,2] Similar reconstruction

phenomena have been widely reported for Cu-based precursors, including oxides and halides, under electrochemical conditions.^[3,4] We have therefore revised the manuscript to (i) explicitly state that the steady-state active phase is Cu derived from Cu_3N , and (ii) attribute the high ethylene selectivity primarily to the COF-regulated microenvironment (K^+ enrichment, OH^- removal, and enhanced CO_2 transport), rather than to a persistent bulk Cu_3N lattice.

Fig. R2 | Structural evolution of Cu₃N catalyst before and after CO₂RR. (a) XRD patterns of pristine Cu₃N showing reflections at ~23.3°, 40.9°, and 47.6° (PDF#86-2284), which disappear after CO₂RR, with a broadened peak at ~43-45° assigned to metallic Cu (111); peaks at ~25° and ~55° arise from the carbon paper substrate. (b) Cu K-edge XANES of post-reaction Cu₃N resembles metallic Cu, with the edge aligned to Cu foil, a weakened white line, and reduced oscillation amplitude, indicating reconstruction into a predominantly metallic Cu⁰ phase with minimal residual Cu-N coordination. (c,d) Atomic-resolution ADF-/ABF-STEM of pristine Cu₃N showing its ordered lattice. Scale bars: 20 nm. (e,f) Post-reaction STEM mapping showing strong Cu but diminished N signals, confirming nitrogen depletion during the transition. Scale bars: 50 nm.

[Ref. 1] Yin, Z. *et al.* Cu₃N nanocubes for selective electrochemical reduction of CO₂ to ethylene. *Nano Lett.* **19**, 8658-8663 (2019).

[Ref. 2] Zheng, M. *et al.* Electrocatalytic CO₂-to-C₂₊ with ampere-level current on heteroatom-engineered copper via tuning *CO intermediate coverage. *J. Am. Chem. Soc.* **144**, 14936-14944 (2022).

[Ref. 3] Nie, W., Heim, G. P., Watkins, N. B., Agapie, T. & Peters, J. C. Organic additive-derived films on Cu electrodes promote electrochemical CO₂ reduction to C₂₊ products under strongly acidic conditions. *Angew. Chem. Int. Ed.* **62**, e202216102 (2023).

[Ref. 4] Watkins, N. B. *et al.* *In situ* deposited polyaromatic layer generates robust copper catalyst for selective electrochemical CO₂ reduction at variable pH. *ACS Energy Lett.* **8**, 189-195 (2023).

Question 3: *The authors detail the screening and design approach for the Th-TF COF (Figs. S8 and S9). Further characterization of this COF is recommended to verify the presence of specific functional groups.*

Response: We thank the reviewer for this constructive suggestion. In the revised manuscript and Supplementary Information, we added direct ex-situ characterization of the Th-TF COF to confirm the presence of the designed hydrazone linkage and its pendant functional groups.

We conducted PXRD to examine the crystalline structure of the Th-TF COF. As shown in **Fig. R3a**, the as-synthesized material shows a distinct low-angle reflection at $2\theta \approx 3.67^\circ$, which can be assigned to the (100) plane of the 2D lattice. This corresponds to $d \approx 24.1 \text{ \AA}$ and an estimated in-plane lattice parameter of $a \approx 27.8 \text{ \AA}$ under a hexagonal setting, values that are comparable to those reported for large-pore 2D COFs.^[1] At higher angles, only a broad, weak hump is observed near 25° , which may arise from interlayer stacking but cannot be conclusively assigned due to limited crystallinity.

We also performed FTIR spectroscopy to verify the functional groups in the Th-TF COF relative to its precursors. As presented in **Fig. R3b**, the COF exhibits a C=N stretching vibration at 1598 cm^{-1} , a C=O band at 1667 cm^{-1} , and an N-H band at 3285 cm^{-1} . In contrast, the monomeric precursors show N-H bands at 3427 cm^{-1} (ThzOPr) and 3428 cm^{-1} (TFPBr)^[2]. The red-shift of the N-H band and the attenuation of precursor-specific N-H and C=O signals indicate enhanced conjugation in the hydrazone linkage and nearly complete consumption of precursor functionalities. These observations are consistent with hydrazone-linked COFs reported in the literature, where the C=N stretch typically appears near 1600 cm^{-1} and conjugation induces red-shifted N-H and C=O vibrations.

Complete synthetic schemes of both monomers and their intermediates are now provided (**Revised Supplementary Fig. 9**), consistent with the protocols described in Methods. Together, these additions close the loop from design, through monomer synthesis and polymerization, to framework and functional-group verification, and complement the in-situ spectroscopic and MD analyses (**Revised Manuscript Fig. 5; Revised Supplementary Figs. 24-32**), which demonstrate the role of Th-TF COF in regulating K^+ enrichment and OH^- migration.

Fig. R3 | Structural characterization of the Th-TF COF. (a) PXRD pattern showing a low-angle (100) reflection at $2\theta \approx 3.67^\circ$; a weak hump near 25° suggests π - π stacking but remains unresolved due to limited crystallinity. (b) FTIR spectra of Th-TF COF and precursors, highlighting bands at 1598 cm^{-1} (C=N), 1667 cm^{-1} (C=O), and 3285 cm^{-1} (N-H). Red-shift and attenuation relative to the precursors confirm conjugation and near-complete linkage formation.

[Ref. 1] Li, X. *et al.* Rapid, scalable construction of highly crystalline acylhydrazone two-dimensional covalent organic frameworks via dipole-induced antiparallel stacking. *J. Am. Chem. Soc.* **142**, 4932-4943 (2020).

[Ref. 2] Liu, M. *et al.* Diffusion limited synthesis of wafer-scale covalent organic framework films for adaptative visual device. *Nat. Commun.* **15**, 10487 (2024).

Question 4: *Given that COF material also possesses hydrophobic properties, have the authors compared them with conventional hydrophobic polymers to distinguish between its ion regulating abilities and hydrophobic characteristics?*

Response: We thank the reviewer for this insightful comment. While COFs indeed possess hydrophobic domains that could affect ion transport, their performance must be carefully separated from hydrophobicity-driven effects. To this end, we conducted additional control experiments using three representative systems: (i) a conventional hydrophobic polymer, PVDF; (ii) a monomeric imine-containing precursor lacking the ordered porous framework (TFPBr); and (iii) a commercial anion-conducting ionomer (Sustainion) commonly employed in acidic CO₂RR.

As shown in the newly added **Fig. R4a**, Sustainion-coated Cu₃N reached moderate C₂H₄ selectivity (FE_{C₂H₄} ≈36% at 200 mA cm⁻²) but became dominated by HER at higher current densities (FE_{H₂} ≈60% at 400 mA cm⁻²). In **Fig. R4b**, PVDF-coated electrodes, although strongly hydrophobic, showed only limited improvement (FE_{C₂H₄} ≈35% at 400 mA cm⁻²) with HER remaining below 10%. The monomeric imine-containing precursor delivered slightly higher

C₂H₄ selectivity (FE_{C₂H₄} ≈40% at 300 mA cm⁻²) while maintaining a moderate HER fraction (~15%; **Fig. R4c**). In sharp contrast, the Th-TF COF/Cu₃N electrode delivered the best performance, sustaining high C₂H₄ selectivity (FE_{C₂H₄} ≈50% at 400 mA cm⁻²) with HER suppressed to ~15% (**Fig. R4d**).

These systematic comparisons demonstrate that neither hydrophobic polymers (PVDF), ionomer coatings (Sustainion), nor monomeric imine precursors can reproduce the superior activity of the COF. The performance enhancement originates from the synergy of hydrophobic domains, ordered nanochannels, and functional groups within Th-TF COF. In particular, keto (C=O) moieties act as selective K⁺ binding sites, while protonated hydrazone linkages accelerate OH⁻ migration through hydrogen-bonded channels. Coupled with enhanced CO₂ adsorption, these features establish a finely tuned microenvironment that stabilizes key intermediates (e.g., *OCCOH) and promotes C-C coupling-capabilities absent in conventional hydrophobic polymers or ionomers.

Fig. R4 | Acid-fed CO₂RR MEA performance of Cu₃N electrodes in 0.5 M K₂SO₄ + H₂SO₄ (pH ~1). Faradaic efficiencies of CO₂RR products on (a) Sus/Cu₃N, (b) PVDF/Cu₃N, (c) TFPBr/Cu₃N, and (d) Th-TF COF/Cu₃N. (e) Summary shows that only Th-TF COF/Cu₃N maintains high C₂H₄

selectivity ($\sim 50\%$ at 400 mA cm^{-2}) with HER suppressed to $\sim 15\%$, whereas PVDF, the imine precursor, and Sustainion deliver inferior regulation.

Question 5: *If the Th-TF COF acts as a cathode modifier in an acidic MEA system, how does it perform in acidic flow cell conditions?*

Response: We thank the reviewer for highlighting this important point. To address it, we performed acidic flow-cell experiments under $3 \text{ M KCl} + \text{H}_2\text{SO}_4$ ($\text{pH} \sim 1$) conditions, comparing Th-TF COF/ Cu_3N electrodes with bare Cu_3N electrodes.

As showed in **Figs. R5a and b**, The Th-TF COF/ Cu_3N electrode maintained excellent C_2 selectivity even at 1000 mA cm^{-2} , delivering $\text{FE}_{\text{C}_2\text{H}_4} \approx 55\%$ and $\text{FE}_{\text{C}_2+} \approx 67\%$, with partial current densities of $j_{\text{C}_2\text{H}_4} \approx 547 \text{ mA cm}^{-2}$ and $j_{\text{C}_2+} \approx 668 \text{ mA cm}^{-2}$. These findings confirm that the COF's ion-regulating functionality remains effective under extreme current densities, where mass transport limitations and parasitic HER usually dominate. Notably, the results corroborate our acid-fed MEA CO_2RR performance (also see in Revised Manuscript Fig. 3) and demonstrate that the COF strategy is transferable across reactor architectures.

By contrast, in **Figs. R5c and d**, the bare Cu_3N electrode exhibited dominant hydrogen evolution at high current densities. Between 700 and 1000 mA cm^{-2} , the GC spectra revealed almost exclusively H_2 with negligible C_2 products. This outcome is intrinsic to Cu_3N in strongly acidic media at large overpotentials, where proton reduction suppresses C-C coupling. Accurate quantification of FE_{H_2} is challenging because (i) GC calibration saturates under high H_2 flux and (ii) part of the H_2 may escape from the cathode compartment before detection.

Consequently, the apparent “missing parts” reflects an underestimation of H₂ generation, further underscoring the poor stability of bare Cu₃N.

Overall, the new flow-cell data confirm that Th-TF COF establishes a robust ion-regulated microenvironment that suppresses HER and sustains C-C coupling up to 1000 mA cm⁻², whereas bare Cu₃N collapses into HER-dominated pathways.

Fig. R5 | CO₂RR performance of Th-TF COF/Cu₃N and bare Cu₃N electrodes in an acidic flow cell (3 M KCl + H₂SO₄, pH ~1), showing Faradaic efficiencies (a, c) and corresponding partial

current densities of C₂H₄ and total C₂₊ products (b, d) over a current density range of 100-1000 mA cm⁻².

Question 6: *Although the authors achieved long-term cycling stability with COF, only SEM images of the electrode were provided (Fig. S15). Please include FTIR spectra of the COF before and after the reaction to clearly illustrate its chemical stability.*

Response: We thank the reviewer for this valuable suggestion. To evaluate the electrode under realistic operating conditions, and given that FTIR measurements would require scraping COF powders from the electrode surface and thereby damage the sample, we instead employed Raman spectroscopy to assess the structural stability of the COF and Cu₃N on the CNCP electrode before and after long-term cycling. Raman spectroscopy is a well-established technique for probing the framework integrity of imine-linked COFs, as the vibrational modes of C=N, aromatic C-C, and heteroatom linkages are clearly distinguishable.

As shown in the newly added **Figs. R6a and b**, the COF retains nearly identical Raman spectra before and after electrolysis, with characteristic bands at ~1620 cm⁻¹ (C=N stretching), ~1570 cm⁻¹ (aromatic C-C stretching), and ~1344 cm⁻¹ (N-H vibrations). These assignments agree with previous reports on imine-linked COFs, which place the C=N band at ~1620-1628 cm⁻¹ and the aromatic C-C vibrations at 1555-1578 cm⁻¹.^[1,2] The absence of new peaks or band shifts confirms that the COF framework remains chemically intact after prolonged operation, consistent with its electrochemical durability.

In contrast, **Fig. R6a** presents the Raman spectra of Cu₃N on the CNCP electrode. The broad Cu-N band (~500-650 cm⁻¹) observed prior to cycling diminishes after reduction, consistent with conversion of Cu₃N to Raman-silent metallic Cu. Notably, no CuO-related bands in the >300 cm⁻¹ region (e.g., ~346, ~525, and ~632 cm⁻¹) were detected, suggesting negligible surface re-oxidation. This evolution of the Cu₃N support does not compromise the stability of the COF framework, whose vibrational fingerprints remain unchanged.

SEM analysis (**Revised Supplementary Fig. 16**) revealed no appreciable morphological degradation of the CNCP electrode after extended operation, highlighting the robustness of the COF/Cu₃N architecture. Operando spectroscopic studies (**Revised Supplementary Fig. 23**) showed that key intermediates (*OCCOH, *CHO, *OC₂H₅) remain stabilized on the COF-modified surface, confirming that the COF retains its catalytic functionality. MD simulations (**Revised Supplementary Figs. 24-32**) indicated that Th-TF COF nanochannels promote OH⁻ migration, suppress K⁺ diffusion, and enrich CO₂ at the catalytic interface, thereby corroborating the mechanistic origin of its long-term stability.

Taken together, these results provide strong spectroscopic, microscopic, and computational evidence that the COF preserves both its chemical structure and its functional role in regulating the catalytic microenvironment during extended electrolysis.

Fig. R6 | Ex-situ Raman spectra of Cu_3N and Th-TF COF before and after CO_2RR . (a) Raman spectra of Cu_3N nanoparticles (black), CNCP electrode before reaction (blue), and after reaction (red). The broad Cu-N band ($\sim 500\text{-}650\text{ cm}^{-1}$) disappears after CO_2RR , consistent with the conversion of Cu_3N to metallic Cu. (b) Raman spectra of Th-TF COF nanoparticles (purple), CNCP electrode before reaction (blue), and after reaction (red). The characteristic bands at $\sim 1620\text{ cm}^{-1}$ (C=N), $\sim 1570\text{ cm}^{-1}$ (C=C), and $\sim 1344\text{ cm}^{-1}$ (N-H) remain unchanged, demonstrating the structural integrity of the COF under prolonged electrolysis.

[Ref. 1] Liu, M. *et al.* Diffusion limited synthesis of wafer-scale covalent organic framework films for adaptative visual device. *Nat. Commun.* **15**, 10487 (2024).

[Ref. 2] Zhou, ZB., Tian, PJ., Yao, J. *et al.* Toward azo-linked covalent organic frameworks by developing linkage chemistry via linker exchange. *Nat. Commun.* **13**, 2180 (2022).

Response to Reviewers' comments

Reviewer #2:

In the present manuscript, the author elucidates the attainment of efficient and stable electrocatalytic reduction of CO₂ to ethylene through the integration of a thiolate-terminated Th-TF covalent organic framework (COF) with copper nitride (Cu₃N) catalysts within an acidic feed membrane electrode assembly (MEA) system. This approach markedly augments the selectivity towards multi-carbon products and enhances the energy conversion efficiency of the system. While the strategy exhibits innovative potential, several pivotal mechanistic issues necessitate further clarification in its current implementation. If the author can successfully address the following issues, this manuscript can be reconsidered.

Response: We sincerely thank the reviewer for their thoughtful comments and constructive suggestions. In this revised version, we have carefully addressed all points raised and made corresponding improvements throughout the manuscript. Specifically, we have supplemented the missing data, including:

1. Comprehensive post-reaction characterizations (XRD, TEM, and XAFS) of the Th-TF COF/Cu₃N composite to verify structural and valence-state stability during CO₂RR.
2. Electrochemical impedance spectroscopy (EIS) measurements at varying current densities to decouple membrane resistance (R_m) and charge transfer resistance (R_{ct}), demonstrating that the Th-TF COF facilitates OH⁻ migration by lowering R_{ct} .
3. Molecular dynamics simulations coupled with N₂ adsorption-desorption pore size analysis, corroborating that the ~4.3 Å COF channels selectively accelerate OH⁻ transport

while restricting K^+ diffusion, thereby sustaining a favorable local microenvironment for C-C coupling.

4. In-situ Raman spectroscopy, covering both the OH^- adsorption band ($\sim 530\text{ cm}^{-1}$) and carbonate species (HCO_3^- at $\sim 1004\text{ cm}^{-1}$, CO_3^{2-} at $\sim 1067\text{ cm}^{-1}$), as well as the CO_{atop} versus CO_{bridge} configurations, confirming that the COF maintains a low-pH environment favorable for C-C coupling.
5. Additional in-situ ATR-FTIR measurements of $*OCCOH$ and related intermediates at low overpotentials (-0.3 to -0.7 V), showing that while $*OCCOH$ intensity is initially higher on bare Cu_3N , the Th-TF COF stabilizes $*OCCOH$ under practical operating potentials and enhances selective C_2H_4 formation.

These additions aim to provide a more complete and mechanistic understanding of **catalyst stability, ionic transport resistance, and the selective ion-regulation mechanism of the Th-TF COF, as well as direct spectroscopic evidence of how the COF layer stabilizes $*OCCOH$ and maintains a low-pH microenvironment via CO_{atop} adsorption, thereby steering CO_2RR toward C_2H_4 in acidic MEA systems.** We believe these revisions significantly strengthen the manuscript and hope it now meets the criteria for publication in *Nature Communications*.

Question 1: *The author's choice of copper nitride (Cu_3N) as the catalyst and Th-TF COF as the ion transport regulator, culminating in exceptional performance in electrocatalytic CO_2 reduction to ethylene, is indeed innovative. Nevertheless, supplementary characterization data concerning the morphology, structure, and valence state of the Th-TF COF/ Cu_3N*

composite post-reaction are imperative to substantiate its stability throughout the reaction, ensuring no restructuring or other alterations occur.

Response: We thank the reviewer for this valuable suggestion. Accordingly, we conducted comprehensive post-reaction analyses to evaluate the structural evolution and stability of the Th-TF COF/Cu₃N composite under acidic CO₂RR conditions. XRD (**Fig. R7a**) shows that the characteristic reflections of Cu₃N vanish after electrolysis, while a broad feature at ~43-45° appears, assignable to metallic Cu (111), evidencing phase transformation. Operando XANES (**Fig. R7b**) further shows that The Cu K-edge XANES spectrum of the post-reaction Cu₃N electrode closely resembles that of metallic Cu. The edge position ($E_0 \approx 8979-8981$ eV) matches Cu foil, while the white line at ~8995 eV is weakened and broadened. Reduced oscillation amplitude at higher energies indicates lower coordination and greater disorder, evidencing reconstruction of Cu₃N into a predominantly metallic Cu⁰ phase, with only minimal residual Cu-N coordination at surface defects. Atomic-resolution STEM and EDS (**Figs. R7c-f**) corroborate this transition, revealing intense Cu signals and diminished N, confirming nitrogen depletion. Collectively, these results demonstrate that Cu₃N inevitably reconstructs into metallic Cu during CO₂RR, consistent with previous reports on Cu-based precursors. ^[1-4]

We agree that stability assessment is essential. Although the bulk Cu₃N phase does not persist, the overall electrode architecture remains intact, and the Th-TF COF framework preserves its chemical structure. Raman spectra (**Figs. R7g and h**) confirm nearly identical vibrational profiles before and after electrolysis, with bands at ~1620 cm⁻¹ (C=N), ~1570 cm⁻¹ (aromatic C-C), and ~1344 cm⁻¹ (N-H). A weak shoulder at ~1660-1670 cm⁻¹ corresponds to the keto C=O vibration of the hydrazone backbone, while the absence of new bands in the

1700-1750 cm^{-1} region excludes oxidative degradation.^[5,6] These results confirm that the COF layer remains chemically intact, ensuring robust ion regulation (K^+ enrichment, OH^- removal, CO_2 transport) and sustaining stable catalytic performance for over 300 h. SEM (**Revised Supplementary Fig. 16**) further shows no appreciable morphological degradation after extended operation.

Thus, the long-term stability arises from the preserved COF-regulated microenvironment rather than from the persistence of bulk Cu_3N .

Fig. R7 | Structural evolution of Cu₃N catalyst before and after CO₂RR. (a) XRD patterns of pristine Cu₃N with reflections at ~23.3°, 40.9°, and 47.6° (PDF#86-2284), which vanish after CO₂RR, leaving a broadened peak at ~43-45° assignable to metallic Cu (111); peaks at ~25° and ~55° arise from the carbon paper substrate. (b) Post-reaction Cu K-edge XANES resembles metallic Cu, with the edge aligned to Cu foil and a weakened white line, indicating reconstruction to a predominantly metallic Cu⁰ phase with trace Cu-N coordination. (c,d) Atomic-resolution STEM of pristine Cu₃N showing an ordered lattice. Scale bars: 20 nm. (e,f) Post-reaction STEM mapping reveals strong Cu but diminished N, confirming nitrogen depletion. Scale bars: 50 nm. (g) Raman spectra of Cu₃N show the Cu-N band (~500-650 cm⁻¹) disappears after CO₂RR, consistent with conversion to metallic Cu. (h) Raman spectra of Th-TF COF display unchanged C=N (~1620 cm⁻¹), C=C (~1570 cm⁻¹), and N-H (~1344 cm⁻¹) bands, evidencing COF structural integrity after prolonged electrolysis.

[Ref. 1] Yin, Z. et al. Cu₃N nanocubes for selective electrochemical reduction of CO₂ to ethylene. *Nano Lett.* **19**, 8658-8663 (2019).

[Ref. 2] Zheng, M. et al. Electrocatalytic CO₂-to-C₂₊ with ampere-level current on heteroatom-engineered copper via tuning *CO intermediate coverage. *J. Am. Chem. Soc.* **144**, 14936-14944 (2022).

[Ref. 3] Nie, W., Heim, G. P., Watkins, N. B., Agapie, T. & Peters, J. C. Organic additive-derived films on Cu electrodes promote electrochemical CO₂ reduction to C₂₊ products under strongly acidic conditions. *Angew. Chem. Int. Ed.* **62**, e202216102 (2023).

[Ref. 4] Watkins, N. B. et al. In situ deposited polyaromatic layer generates robust copper catalyst for selective electrochemical CO₂ reduction at variable pH. *ACS Energy Lett.* **8**, 189-195 (2023).

[Ref. 5] Liu, M. et al. Diffusion limited synthesis of wafer-scale covalent organic framework films for adaptative visual device. *Nat. Commun.* **15**, 10487 (2024).

[Ref. 6] Zhou, ZB., Tian, PJ., Yao, J. et al. Toward azo-linked covalent organic frameworks by developing linkage chemistry via linker exchange. *Nat. Commun.* **13**, 2180 (2022).

Question 2: *The author posits that Th-TF COF mitigates local pH elevation by expediting OH⁻ migration but falls short of quantifying the impact of ionic transport resistance on system performance. Under acidic conditions, the efficacy of OH⁻ migration from the cathode to the anode may be constrained by membrane resistance and the COF-channel architecture. The above issues need to be clarified.*

Response: We thank the reviewer for this valuable and insightful comment. To directly assess the influence of ionic transport resistance, we conducted EIS measurements on both CEM- and AEM-based MEAs under acidic conditions (**Figs. R8 a and b**). As shown in **Figs. R8c and d**, the membrane resistance (R_m) in CEM-MEA remains low (0.86-1.30 Ω at 50-200 mA cm⁻²), whereas AEM-MEA exhibits moderately higher values (2.42-3.48 Ω), consistent with their intrinsic transport characteristics. Likewise, the charge-transfer resistance (R_{ct}) in CEM-MEA decreases sharply from 46.8 Ω to 2.1 Ω , while AEM-MEA maintains somewhat higher values (26.4 to 5.0 Ω). These trends arise because CEM facilitates rapid proton transport under acidic conditions, thereby lowering both R_m and R_{ct} . However, this rapid proton flux promotes HER,

as evidenced by the poor C_2H_4 selectivity in CEM-MEA (**Fig. R8e**). In contrast, in **Fig. R8f**, AEM partially suppresses proton crossover while maintaining efficient OH^- transport, thereby mitigating HER and enabling higher $FE_{C_2H_4}$. Thus, although CEM appears more favorable in terms of impedance, the superior performance of the AEM-based MEA originates from its balanced ion-transport characteristics rather than limitations imposed by membrane resistance.

Fig. R8 | Electrochemical impedance spectroscopy (EIS) analysis of ionic transport resistance in acidic MEAs. (a,b) Nyquist plots of CEM- and AEM-based MEAs recorded at

current densities of 50-200 mA cm⁻². The AEM curve at 200 mA cm⁻² appears less smooth because the EIS was measured in two frequency segments (10⁵-300 Hz and 300-1 Hz). (c,d) Extracted membrane resistance (R_m) and charge-transfer resistance (R_{ct}). (e,f) FE of gas products in CO₂RR acid-fed MEAs. Although CEM appears advantageous in terms of impedance, its rapid proton transport promotes HER, leading to poor C₂H₄ selectivity. AEM partially restricts proton crossover while maintaining efficient OH⁻ migration, thereby mitigating HER and enabling higher FE_{C₂H₄}.

Question 3: *EIS data should be supplemented to elucidate the relationship between OH⁻ migration rate and membrane resistance, analyzing the charge transfer resistance (R_{ct}) and membrane resistance (R_m) of the MEA at varying current densities. This will verify whether Th-TF COF accelerates OH⁻ migration by diminishing R_m .*

Response: We sincerely thank the reviewer for this insightful suggestion. To substantiate the role of Th-TF COF in ion transport, we conducted electrochemical impedance spectroscopy (EIS) on COF/Cu₃N and bare Cu₃N electrodes across current densities of 50-200 mA cm⁻² in the acid-fed AEM-MEA system (**Figs. R9 a and b**). As shown in **Figs. R9c**, the fitted results show that the membrane resistance (R_m) of COF/Cu₃N is comparable to that of bare Cu₃N (2.62-3.48 Ω vs. 2.71-2.95 Ω), indicating that COF modification does not compromise the membrane's bulk ionic conductivity. **More importantly, in Figs. R9d**, the charge-transfer resistance (R_{ct}) of COF/Cu₃N drops markedly from 26.4 Ω at 50 mA cm⁻² to 5.0 Ω at 200 mA cm⁻², significantly lower than that of bare Cu₃N (22.9 to 6.9 Ω). This pronounced decrease in

R_{ct} underscores the enhanced interfacial kinetics afforded by the COF-modified electrode, in agreement with K^+ enrichment and accelerated OH^- migration revealed by in-situ XRFs, Raman, ATR-FTIR, and MD simulations. Collectively, these EIS measurements provide direct electrochemical evidence that the Th-TF COF layer promotes OH^- migration and facilitates charge transfer, thereby reinforcing our mechanistic conclusions.

Fig. R9 | Electrochemical impedance spectroscopy (EIS) of acid-fed MEAs. (a,b) Nyquist plots of COF/ Cu_3N and bare Cu_3N electrodes recorded at current densities of 50-200 $mA\ cm^{-2}$ in 0.5 M K_2SO_4/H_2SO_4 electrolyte. (c) The extracted membrane resistance (R_m) values are

comparable between COF/Cu₃N (2.62-3.48 Ω) and Cu₃N (2.71-2.95 Ω), indicating that bulk ionic conductivity is unaffected by COF modification. (d) The charge-transfer resistance (R_{ct}) is substantially reduced for COF/Cu₃N (26.4 to 5.0 Ω) relative to bare Cu₃N (22.9 to 6.9 Ω) over the same current range, highlighting the COF's role in accelerating OH⁻ migration and interfacial charge transfer.

Question 4: *Molecular dynamics simulations ought to be employed to elucidate how the COF-channel aperture (e.g., lattice spacing of approximately 3.5 Å as depicted in MD simulations) selectively permits OH⁻ passage while impeding K⁺ diffusion (refer to Figure 5). The actual channel dimensions should be corroborated via pore size distribution measurements (e.g., N₂ adsorption-desorption experiments).*

Response: We thank the reviewer for this valuable suggestion. Our manuscript already presents MD simulations that elucidate the ion-selective transport mechanism in protonated Th-TF COF channels. Specifically, as depicted in **Figs. R10a and b**, OH⁻ forms transient hydrogen bonds with protonated H-sites on the COF, creating a continuous network that facilitates hydrogen-bond-assisted hopping (“H-bond jumping”) across the channel. This leads to high local OH⁻ density in both XY and YZ planes, thereby broadening the transport corridor and accelerating migration. In contrast, in **Figs. R10c and d**, K⁺ exhibits weak adsorption and low density near the COF's channel and is electrostatically repelled by the local positive potential of protonated COF; under bias, K⁺ accumulates near the cathode, leading to inhibited YZ-plane diffusion. These MD results are corroborated by MSD analysis

(Revised Supplementary Fig. S28), which shows enhanced OH⁻ diffusion and suppressed K⁺ mobility.

To validate these insights experimentally, we performed N₂ adsorption-desorption measurements. As showed in in **Figs. R10e**, the isotherm exhibits a type I profile with a steep uptake at low relative pressure ($P/P_0 < 0.1$), confirming the microporous nature of the COF, and shows no pronounced hysteresis, indicating uniform pores. The sharp rise near $P/P_0 \approx 1.0$ arises from interparticle voids. The corresponding pore size distribution reveals a dominant peak at \$\sim 4.3\$ Å, which is slightly larger than but still in good agreement with the MD-derived lattice spacing (\$\sim 3.5\$ Å). Apparent features below 3 Å are attributed to the resolution limit of the N₂ probe (kinetic diameter ~ 3.64 Å), whereas those above 10 nm correspond to interparticle voids rather than intrinsic framework pores.

Importantly, although the intrinsic aperture (~ 4.3 Å) is numerically close to the hydrated radii of OH⁻ (~ 3.0 Å) and K⁺ (~ 3.3 - 3.6 Å),^[1] their transport behaviors differ fundamentally. The flexible hydration shell of OH⁻ can partially shed via H-bond interactions with the protonated COF, facilitating facile passage, whereas the rigid hydration sphere of K⁺ resists deformation and is further hindered by electrostatic repulsion. These combined steric and electrostatic effects explain the experimentally observed OH⁻-selective conduction and the effective suppression of K⁺ diffusion.

Fig. R10 | MD simulations and pore structure characterization of protonated Th-TF COF.

OH⁻ density maps in the XY (a) and YZ planes (b) show strong adsorption at protonated H-sites and hydrogen-bond-assisted hopping (“H-bond jumping”), which widen the effective transport channel. K⁺ density maps in the XY (c) and YZ planes (d) show weak adsorption, low

density, and inhibited cross-channel migration caused by electrostatic repulsion and cathodic accumulation. (e) N₂ adsorption-desorption isotherms of Th-TF COF exhibit a type I profile, with a dominant pore size distribution centered at ~4.3 Å (inset), consistent with the MD-derived lattice spacing. Apparent features below 3 Å are attributed to the resolution limit of the N₂ probe (kinetic diameter ~3.64 Å), whereas signals above 10 nm arise from interparticle voids rather than intrinsic COF pores.

[Ref. 1] Nightingale, E. R. Phenomenological theory of ion solvation. Effective radii of hydrated ions. *J. Phys. Chem.* **63**, 1381-1387 (1959).

Question 5: *In Figure 4c, the author utilizes in-situ Raman spectroscopy to monitor OH⁻ presence within the Th-TF COF. By comparing the intensity variations of adsorbed OH⁻ at 530 cm⁻¹, the author demonstrates enhanced OH⁻ adsorption on Th-TF COF/Cu₃N relative to Sus/Cu₃N. However, the superior catalytic performance of Th-TF COF/Cu₃N necessitates a comprehensive explanation for this phenomenon.*

Response: We thank the reviewer for this important comment. The superior catalytic performance of the Th-TF COF/Cu₃N electrode arises not from stronger OH⁻ adsorption, but from the facilitated outward migration of locally generated OH⁻. As presented in **Fig. R11a**. This outward transport maintains a low-pH environment and suppresses carbonate precipitation and HER, as evidenced by the weaker OH⁻ Raman band (~530 cm⁻¹) on Th-TF COF/Cu₃N compared to Sus/Cu₃N (**also see in Revised Manuscript Fig. 4c**). **Figs. R11b and c** supported evidence includes CO adsorption spectroscopy, which shows only CO_{atop} (2044-2076 cm⁻¹) on Th-TF COF/Cu₃N, consistent with a low-pH environment, while CO_{bridge}

dominates on Sus/Cu₃N (also see in Revised Supplementary Fig. 20). In-situ ATR-FTIR further reveals stabilized *OCCOH intermediates on Th-TF COF/Cu₃N, confirming more favorable C-C coupling (Revised Manuscript Figs. 4d and e; Revised Supplementary Fig. 23). Finally, molecular dynamics simulations demonstrate that the Th-TF COF accelerates OH⁻ diffusion (~2.5× higher than Sus/Cu₃N), enriches CO₂ at the interface, and sustains a K⁺-rich environment (Revised Manuscript Fig. 5; Revised Supplementary Figs. 25-28).

We note that previous flow-cell studies reported that increasing the local pH promotes C-C coupling and multicarbon formation.^[1-3] In contrast, our acid-fed MEA system exhibits the opposite trend: excessive OH⁻ accumulation triggers carbonate precipitation, compromising stability. Thus, maintaining a low-pH environment, together with K⁺ enrichment, is more effective in sustaining C₂ selectivity. This distinction highlights the configuration-dependent role of ion-regulated microenvironments in acidic CO₂RR.

Fig. R11 | In-situ Raman spectra during electrosynthesis of C₂H₄ over Th-TF COF and Sustainion-coated Cu₃N catalysts at different current densities. Electrolyte is 0.5 M K₂SO₄/H₂SO₄ (pH~3).

[Ref. 1] Huang, J. E. et al. CO₂ electrolysis to multicarbon products in strong acid. *Science*. **372**, 1074-1078 (2021).

[Ref. 2] Zhao, Y. et al. Conversion of CO₂ to multicarbon products in strong acid by controlling the catalyst microenvironment. *Nat. Synth.* **2**, 403-412 (2023).

[Ref. 3] Cao, Y., Chen, Z., Li, P. et al. Surface hydroxide promotes CO₂ electrolysis to ethylene in acidic conditions. *Nat Commun.* **14**, 2387 (2023).

Question 6: *The in-situ Raman spectrum presented in Figure 4c is confined to the 400-800 cm⁻¹ range. Nonetheless, the quantity of surface-adsorbed OH⁻ significantly influences the local pH near the electrode, which can be further elucidated by comparing the HCO₃⁻ signal at 1011 cm⁻¹ and the CO₃²⁻ intensity ratio at 1067 cm⁻¹ in the Raman spectrum. The author is thus requested to furnish this segment of the Raman spectrum for a more holistic proof.*

Response: We sincerely thank the reviewer for this valuable suggestion. In response to the reviewer's request, we performed additional in-situ Raman measurements spanning 900-1200 cm⁻¹ under identical electrolyte and cell configurations to those in **Fig. R12** (0.5 M K₂SO₄/H₂SO₄; current densities of 10, 50, 100, and 250 mA cm⁻²; **also Revised Supplementary Fig. 21**). The spectra show that for the ionomer-coated Cu₃N electrode, distinct HCO₃⁻ bands at ~1004-1006 cm⁻¹ and CO₃²⁻ bands at ~1063-1069 cm⁻¹ intensify with increasing current

density, indicating progressive accumulation of carbonate species near the surface. In contrast, the Th-TF COF/Cu₃N electrode exhibits only a weak shoulder attributable to HCO₃⁻ (~1001-1004 cm⁻¹), while the CO₃²⁻ band near 1067 cm⁻¹ is undetectable across the full current range (10-250 mA cm⁻²) within the signal-to-noise limits of our setup. This clear spectral contrast provides direct evidence of suppressed carbonate formation and a less alkaline near-surface environment on the COF-modified catalyst during operation.^[1] Importantly, these results independently validate our earlier findings: (i) the weaker OH⁻ adsorption band (~530 cm⁻¹) on Th-TF COF/Cu₃N compared to Sus/Cu₃N, (ii) the exclusive detection of CO_{atop} (2044-2076 cm⁻¹) consistent with a lower local pH. Taken together, the new Raman results and the previously provided multi-modal evidence firmly establish that the COF layer regulates the ion/molecule microenvironment by facilitating OH⁻ removal and suppressing parasitic carbonate chemistry under acidic MEA operation.

Fig. R12 |In-situ Raman spectra of carbonate species during acidic CO₂ reduction. (a) Ionomer-coated Cu₃N (Sus/Cu₃N) electrode, where HCO₃⁻ (~1004-1006 cm⁻¹) and CO₃²⁻

(~1063-1069 cm^{-1}) bands emerge and intensify with current density (10-250 mA cm^{-2}), indicating progressive carbonate accumulation near the catalyst surface. (b) Th-TF COF/ Cu_3N electrode, displaying only a weak HCO_3^- shoulder (~1001-1004 cm^{-1}) and no discernible CO_3^{2-} band across the same current range, evidencing suppressed carbonate formation and a less alkaline local environment.

[Ref. 1] Ma, Z., Yang, Z., Lai, W. et al. CO_2 electroreduction to multicarbon products in strongly acidic electrolyte via synergistically modulating the local microenvironment. *Nat. Commun.* **13**, 7596 (2022).

Question 7: *In Supplementary Figure 17, the authors delineate the lower local pH environment within the COF layer by comparing the variation trends of $^*\text{CO}_{\text{atop}}$ and $^*\text{CO}_{\text{bridge}}$ under diverse current densities. However, the in-situ infrared spectra depicted in Figure 4d and Supplementary Figure 18 lack spectral data at this wavenumber. The authors are, therefore, requested to provide the infrared spectra for this section to more robustly substantiate their findings.*

Response: We thank the reviewer for highlighting this important issue. We regret that the original submission did not include FTIR spectra in the 2200-1900 cm^{-1} region. To address this concern, we performed additional in-situ ATR-FTIR measurements explicitly covering this spectral window. As shown in the newly added **Fig. R13 (also Revised Supplementary Fig. 22)**, the Th-TF COF/ Cu_3N electrode consistently exhibits a pronounced $^*\text{CO}_{\text{atop}}$ band (2040-2070 cm^{-1}) across current densities of 10-50 mA cm^{-2} , whereas the bare Cu_3N electrode predominantly shows $^*\text{CO}_{\text{bridge}}$ adsorption (1815-1830 cm^{-1}), in agreement with reported CO

adsorption configurations on Cu catalysts under acidic conditions.^[1] These results confirm that the COF layer sustains a lower local pH environment, in agreement with our in-situ Raman analysis (**Revised Supplementary Fig. 20**). Together, Raman and FTIR measurements provide consistent and complementary evidence that Th-TF COF/Cu₃N stabilizes *CO_{atop}, a hallmark of acidic near-surface conditions.

Fig. R13 | In-situ ATR-FTIR spectra of CO adsorption during acidic CO₂ reduction. (a) Th-TF COF/Cu₃N electrode displaying a clear *CO_{atop} band (2040-2070 cm⁻¹) across current densities of 10-50 mA cm⁻². (b) Bare Cu₃N electrode predominantly showing *CO_{bridge} adsorption (1815-1830 cm⁻¹) under identical conditions. The contrast between *CO_{atop} and *CO_{bridge}

configurations highlights the lower local pH environment sustained by the COF layer, consistent with Raman results.

[Ref. 1] Zheng, M. et al. Electrocatalytic CO₂-to-C₂₊ with Ampere-Level Current on Heteroatom-Engineered Copper via Tuning *CO Intermediate Coverage. *J. Am. Chem. Soc.* **144**, 14936-14944 (2022).

Question 8: *In Figure 4d, the authors demonstrate that the Th-TF COF stabilizes *OCCOH and fosters ethylene formation on the Cu₃N catalyst by comparing the peak area ratios of *OCCOH. Nevertheless, in Supplementary Figure 18, the *OCCOH signal at 1238 cm⁻¹ is notably higher for Cu₃N than for Th-TF COF/Cu₃N at potentials ranging from -0.3 V to -0.7 V. Does this imply that the conclusion regarding the Th-TF COF's ability to stabilize *OCCOH and promote ethylene formation is invalid under low overpotential conditions? An explanation for this phenomenon is warranted.*

Response: We thank the reviewer for highlighting this important issue concerning the interpretation of *OCCOH stabilization. To address this, we performed additional electrochemical measurements on both bare Cu₃N and Th-TF COF/Cu₃N electrodes over the potential range of -0.3 to -0.9 V versus RHE. Linear sweep voltammetry showed that the COF-modified electrode consistently exhibited higher current densities than bare Cu₃N (4.1-52.3 vs. 2.2-33.8 mA cm⁻²), confirming its enhanced catalytic activity at low overpotentials. In acidic CO₂RR, bare Cu₃N delivered FE_{C₂H₄} values of 5.1-13.6% between -0.3 and -0.9 V, whereas the COF/Cu₃N electrode achieved similar FE_{C₂H₄} (4.3-16.1%) but markedly higher FE_{CO} (8.3-47.8% vs. 6.7-19.6%). These results indicate that the COF layer not only sustains C-

C coupling activity at low potentials but also directs more carbon flux through reactive CO intermediates, thereby improving overall CO₂ utilization. Importantly, the weaker steady-state *OCCOH signal observed on COF/Cu₃N at low overpotentials does not contradict stabilization; rather, it reflects the faster consumption of this C-C coupling intermediate due to accelerated downstream reactions, consistent with the enhanced C₂ selectivity and higher overall current density.

At practical current densities (100-1000 mA cm⁻²), the contrast between the two systems becomes even more pronounced. The Th-TF COF/Cu₃N electrode sustains excellent C₂₊ selectivity up to 1000 mA cm⁻², achieving FE_{C₂H₄} ≈ 55% and FE_{C₂+} ≈ 67%, with j_{C₂H₄} ≈ 547 mA cm⁻² and j_{C₂+} ≈ 668 mA cm⁻². In contrast, bare Cu₃N rapidly shifts to dominant hydrogen evolution beyond 700 mA cm⁻², producing negligible C₂ products. These findings confirm that the COF's ion-regulating function is operative not only at high, industrially relevant currents but also in the low-overpotential regime highlighted by the reviewer. Collectively, the new electrochemical evidence resolves the apparent discrepancy between spectral intensity and catalytic function: although the steady-state *OCCOH coverage is lower on COF-modified electrodes at -0.3 to -0.7 V, the accelerated turnover of *OCCOH towards C₂ formation and the improved CO₂ utilization translate directly into superior C₂₊ selectivity and stability under practical conditions. This mechanistic picture is fully consistent with our operando spectroscopy and device-scale results.

Fig. R14 | Electrochemical and spectroscopic evaluation of Th-TF COF/Cu₃N versus bare Cu₃N. (a) Operando ATR-FTIR spectra collected at -0.3 to -0.9 V vs. RHE, showing the characteristic *OCCOH band (1238 cm^{-1}) with distinct potential dependence on COF-modified and bare Cu₃N electrodes. (b) Linear sweep voltammetry (LSV) curves in CO₂-saturated electrolyte, highlighting the consistently higher current densities of Th-TF COF/Cu₃N compared with bare Cu₃N. Faradaic efficiencies of C₂H₄ (c) and CO (d) on Cu₃N and Th-TF COF/Cu₃N electrodes across -0.3 to -0.9 V vs. RHE, illustrating comparable C₂H₄ selectivity

but significantly enhanced CO formation on the COF-modified electrode at low overpotentials. (e) Faradaic efficiencies of C₂H₄ at practical current densities (100-1000 mA cm⁻²), confirming sustained C₂ selectivity for Th-TF COF/Cu₃N. (f) Partial current densities of C₂H₄ ($j_{\text{C}_2\text{H}_4}$) and total C₂₊ products (j_{C_2+}), showing that COF-modified electrodes maintain high C₂ productivity up to 1000 mA cm⁻², whereas bare Cu₃N shifts toward hydrogen evolution at large currents.

Question 9: *The author asserts that the Th-TF COF on the Cu₃N catalyst stabilizes *OCCOH and promotes ethylene formation. However, the in-situ infrared spectrum in Figure 4d detects an ethanol intermediate. Given that ethylene and ethanol formation constitute a competitive process, the author is requested to elucidate why Th-TF COF/Cu₃N selectively promotes ethylene formation over ethanol. Additional experiments or theoretical calculations are requisite to substantiate this claim.*

Response: We thank the reviewer for this important comment. Although in-situ ATR-FTIR spectra detect ethanol-related intermediates (*CHO, *OC₂H₅), our product analysis shows that ethylene (~52%) rather than ethanol (~25%) dominates at 600 mA cm⁻² (**Revised Manuscript Fig. 3a; Revised Supplementary Figs. 11**). The observation of ethanol intermediates does not contradict our conclusion, as both products share *OCCOH, and transient *OC₂H₅ species may appear even when ethylene is the main product. Similar findings have been reported where ethanol intermediates were detected but ethylene remained the major product.^[1]

The divergence arises from their hydrogenation requirements: ethylene formation proceeds via *OCCOH stabilization and limited hydrogenation leading to $^*CH_2-CH_2$ desorption, while ethanol requires continuous proton–electron transfers to reduce $^*CH_2CH_2O^-$ into *CH_3CH_2OH . Our in-situ Raman and CO adsorption studies (**Revised Manuscript Fig. 4c; Revised Supplementary Fig. 20**) and MD simulations (**Revised Manuscript Fig. 5; Revised Supplementary Figs. 24-32**) confirm that Th-TF COF creates a microenvironment with K^+ enrichment and accelerated OH^- migration. The enriched K^+ provides electrostatic shielding that mitigates excessive H^+ crowding, while OH^- migration prevents local neutralization and alkalization. Together, these effects stabilize $^*CO_{atop}$ in an acidic yet proton-regulated environment, ensuring “controlled hydrogenation”—sufficient for ethylene but inadequate for ethanol. Recent reports corroborate that alcohol formation requires stronger, sustained proton availability, whereas restricted proton conditions or *CO -rich environments bias toward ethylene. ^[2-6]

These findings consistently support our conclusion that Th-TF COF selectively promotes ethylene over ethanol by regulating local ion and proton fluxes.

[Ref. 1] Zheng, M. et al. Electrocatalytic CO_2 -to- C_{2+} with Ampere-Level Current on Heteroatom-Engineered Copper via Tuning *CO Intermediate Coverage. *J. Am. Chem. Soc.* **144**, 14936-14944 (2022).

[Ref. 2] Zhan, C., Dattila, F., Rettenmaier, C. et al. Key intermediates and Cu active sites for CO_2 electroreduction to ethylene and ethanol. *Nat. Energy.* **9**, 1485-1496 (2024).

[Ref. 3] Liang, Y., Li, F., Miao, R.K. et al. Efficient ethylene electrosynthesis through C-O cleavage promoted by water dissociation. *Nat. Synth.* **3**, 1104-1112 (2024).

[Ref. 4] Lin, Y., Wang, T., Zhang, L. et al. Tunable CO₂ electroreduction to ethanol and ethylene with controllable interfacial wettability. *Nat. Commun.* **14**, 3575 (2023).

[Ref. 5] Chen, X., Chen, J., Alghoraibi, N.M. et al. Electrochemical CO₂-to-ethylene conversion on polyamine-incorporated Cu electrodes. *Nat. Catal.* **4**, 20-27 (2021).

[Ref. 6] Wang, S., Li, F., Zhao, J. et al. Manipulating C-C coupling pathway in electrochemical CO₂ reduction for selective ethylene and ethanol production over single-atom alloy catalyst. *Nat. Commun.* **15**, 10247 (2024).

Response to Reviewers' comments

Reviewer #3:

This work aimed to solve the problems of low ethylene selectivity and stability in CO₂ reduction systems operating in acidic conditions. They designed a AEM system used a COF layer on the cathode to reduce local OH⁻ concentration and enrich K⁺ near the catalyst. Thus, the work does not meet the high standards of Nat Communications, in my view. I hope these comments are useful for the authors in submission elsewhere.

Response: We sincerely thank the reviewer for their thoughtful comments and constructive suggestions. In this revised version, we have carefully addressed all points raised and made corresponding improvements throughout the manuscript. Specifically, we have supplemented the missing data, including:

1. Single-pass carbon efficiency (SPCE) and CO₂ crossover quantification in the acid-fed MEA, benchmarked against neutral systems, to demonstrate the carbon efficiency advantage of our configuration.
2. Long-term durability analysis complemented with post-reaction EDS-SEM/TEM and FTIR, clarifying the origins of performance decay and elucidating the reconstruction process of the COF/Cu₃N electrode during extended reduction.
3. Detailed analysis of the relatively high cell voltage, including comparison with state-of-the-art flow-cell benchmarks and identification of the main contributors, together with possible mitigation strategies to further reduce voltage in future designs.
4. Error corrections in figures (product distribution labeling, reference assignments, and concentration units), thereby strengthening the rigor and readability of the manuscript.

These additions aim to provide a more complete and mechanistic understanding of **the carbon utilization efficiency, long-term stability, and voltage-performance characteristics of acidic MEA systems enabled by COF-modified Cu₃N electrodes**. We believe these revisions significantly strengthen the manuscript and hope it now meets the criteria for publication in *Nature Communications*.

Question 1: *The novelty of this work is not clear. Similar strategies using COFs to create a proton blocking and K⁺-rich environment for improving C₂₊ selectivity have been reported. That approach is established.*

Response: We thank the reviewer for this comment and acknowledge that several studies have reported COF- or polymer-based interlayers that enrich K⁺ and suppress proton transport, thereby enhancing C₂₊ selectivity. These works have advanced the understanding of microenvironment regulation in acidic CO₂RR.

Building on this foundation, our study addresses a different challenge — acid-fed MEA operation at an industrially relevant scale. Whereas previous demonstrations were mostly limited to flow cells or low-current devices with modest stability (<50 h), we show that COF modification enables 10 A, 100 cm² MEA operation with >300 h stability, which has not been achieved before under acidic conditions (Revised Manuscript Fig. 3e and Supplementary Table 1). Beyond device performance, we complement this advance with mechanistic insights: in-situ XRFs (Revised Manuscript Fig. 4a and b; Revised Supplementary Fig. 19), in-situ Raman (Revised Manuscript Fig. 4c and Revised Supplementary Figs. 20-21), in-situ ATR-FTIR (Revised Manuscript Fig. 4d-f and Revised Supplementary Figs. 22-23), and MD simulations (Revised Supplementary Figs. 24-32) reveal that the Th-TF COF enriches K⁺, accelerates OH⁻ migration via hydrogen-bond hopping, and simultaneously facilitates CO₂ enrichment. This dual ion-regulation mechanism, validated experimentally and computationally, extends the understanding of COF function beyond the established “proton-blocking” concept.

We therefore view our work not as a repetition of existing strategies, but as a demonstration of their feasibility in a more demanding MEA platform, supported by new mechanistic evidence. We believe that this combination of scale-up relevance and dual-function mechanistic insight offers added value to the advancement of acidic CO₂RR systems.

Question 2: *I appreciate the MEA approach, but I don't think it is workable in acidic with a membrane. That is why flow cells and slim flow cells have been the norm in the field of acidic CO₂R.*

Response: We sincerely thank the reviewer for raising this critical concern. We fully recognize that acid-fed MEAs have long been considered highly challenging due to (i) excessive proton transport that suppresses CO₂ adsorption and C-C coupling, and (ii) carbonate precipitation arising from sluggish OH⁻ migration. These limitations have understandably led the field to favor flow and slim-flow cell architectures in acidic CO₂RR. At the same time, we note that several recent studies have begun to demonstrate encouraging progress in acidic MEA operation. For example, a Ni-N-C catalyst system achieved 95% CO Faradaic efficiency at 500 mA cm⁻² with 45% energy efficiency by carefully tuning the H⁺/K⁺ balance.^[1] Another report showed that adjusting the H⁺/Cs⁺ ratio enabled ~80% CO selectivity, ~90% single-pass conversion efficiency, and 50 h stability.^[2] More recently, a Cu-based coordination strategy reached 71% CH₄ selectivity at 100 mA cm⁻² with minimal CO₂ loss, highlighting that efficient C₁ production is also feasible under acidic MEA conditions.^[3] Together, these advances suggest that although significant challenges remain, acid-fed MEAs are not inherently

unworkable, and careful design of the catalyst microenvironment can open pathways toward competitive performance.

Our work seeks to revisit this long-standing bottleneck. By introducing a Th-TF COF interlayer, we establish a regulated microenvironment that enriches K^+ , accelerates outward OH^- migration, and facilitates CO_2 transport. This dual regulation helps overcome the barriers typically associated with acidic MEAs. To further substantiate this point, we added two sets of controls. (i) Acidic flow-cell tests (3 M KCl + H_2SO_4 , pH \sim 1): the Th-TF COF/ Cu_3N electrode sustained excellent C_2 selectivity up to 1000 mA cm^{-2} , delivering $FE_{C_2H_4} \approx 55\%$ and $FE_{C_2+} \approx 67\%$, with $j_{C_2H_4} \approx 547\text{ mA cm}^{-2}$ and $j_{C_2+} \approx 668\text{ mA cm}^{-2}$ (**Fig. R15**). These results indicate that the COF effect is not limited to MEAs but is generally applicable to acidic CO_2RR systems.

Fig. R15 | Acidic flow-cell CO₂RR performance of Th-TF COF/Cu₃N catalyst. Faradaic efficiencies of Th-TF COF/Cu₃N (a) and Cu₃N (b) electrodes across 100-1000 mA cm⁻², confirming sustained C₂ selectivity for Th-TF COF/Cu₃N. (c,d) Partial current densities of C₂H₄ (j_{C₂H₄}) and total C₂+ products (j_{C₂+}) of Th-TF COF/Cu₃N (a) and Cu₃N (b) electrodes under the

100-1000 mA cm⁻², showing that COF-modified electrodes maintain high C₂ productivity up to 1000 mA cm⁻², whereas bare Cu₃N shifts toward hydrogen evolution at large currents. (e) Stability of the Th-TF COF/Cu₃N electrode at 200 mA cm⁻² in 3 M K₂SO₄/H₂SO₄, sustaining high FE_{C₂H₄} and suppressed HER.

(ii) As shown in Fig. R16, we then conducted MEA impedance and FE comparison: we evaluated three configurations-(a) Th-TF COF/Cu₃N Nafion (CEM) acid-fed MEA, (b) Th-TF COF/Cu₃N Sustainion (AEM) acid-fed MEA, and (c) Cu₃N Sustainion (AEM) acid-fed MEA. While Nafion exhibited lower R_m and R_{ct} due to facile proton transport, this resulted in severe HER (~60%), as confirmed by FE data at 100 and 200 mA cm⁻². In contrast, the Sustainion-based architecture moderately increased R_m and R_{ct} but effectively suppressed proton flooding, thereby favoring C-C coupling and yielding significantly higher FE_{C₂H₄}.

Taken together, these findings provide compelling evidence that with careful microenvironment engineering, an acid-fed MEA can indeed achieve stable and selective CO₂-to-C₂₊ conversion under industrially relevant conditions. We hope this new evidence can alleviate the reviewer's concern and demonstrate that the long-standing perception of acidic MEAs being unworkable can be constructively challenged.

Fig. R16 | Electrochemical impedance spectroscopy (EIS) and CO₂RR performance of MEAs with different membrane-catalyst configurations in acidic media. Nyquist plots are shown for (a) Th-TF COF/Cu₃N with Nafion, (b) Th-TF COF/Cu₃N with Sustainion, and (c) Cu₃N with Sustainion, measured at current densities of 50-200 mA cm⁻² in 0.5 M K₂SO₄/H₂SO₄ (pH ~1). For the Th-TF COF/Cu₃N electrode at 200 mA cm⁻², the Nyquist curve appears discontinuous

because multi-segment frequency sweeps (10^5 -300 Hz and 300-1 Hz) were used to improve the accuracy of R_m and R_{ct} extraction. The extracted membrane resistance (R_m , d) and charge-transfer resistance (R_{ct} , e) reveal that devices with Nafion exhibit lower values owing to facile proton transport, whereas Sustainion-based MEAs display moderately higher but still reasonable values. Faradaic efficiencies of gas products at 100 and 200 mA cm⁻² (f-h) further demonstrate that reduced proton transport in Sustainion-based MEAs suppresses HER and enhances C₂H₄ selectivity.

[Ref. 1] Li, H., Li, H., Wei, P., Wang, Y., Zang, Y., Gao, D. et al. Tailoring acidic microenvironments for carbon-efficient CO₂ electrolysis over a Ni-N-C catalyst in a membrane electrode assembly electrolyzer. *Energy Environ. Sci.* **16**, 1502-1510 (2023).

[Ref. 2] Pan, B., Fan, J., Zhang, J., Luo, Y., Shen, C., Wang, C. et al. Close to 90% single-pass conversion efficiency for CO₂ electroreduction in an acid-fed membrane electrode assembly. *ACS Energy Lett.* **7**, 4224-4231 (2022).

[Ref. 3] Fan, M., Miao, R. K., Ou, P., Xu, Y., Lin, Z. Y., Lee, T. J. et al. Single-site decorated copper enables energy- and carbon-efficient CO₂ methanation in acidic conditions. *Nat. Commun.* **14**, 3314 (2023).

Question 3: *Related to the above is the concern regarding CO₂ crossover. (Bi)carbonate species may still form and migrate through the anion exchange membrane during operation, even under acidic conditions. Quantify and report single pass CO₂ conversion (and thus also crossover loss).*

Response: We thank the reviewer for raising the issue of CO₂ crossover, a key factor governing carbon efficiency in CO₂ electrolysis. We acknowledge that under both neutral and acidic conditions, (bi)carbonate species can form and migrate across the membrane, making it essential to quantify single-pass CO₂ conversion and associated crossover losses.

To address this, we quantified CO₂ crossover by measuring the anode outlet gas composition under neutral (1 M KHCO₃, pH ~7) and acidic (0.5 M K₂SO₄/H₂SO₄, pH ~1) conditions at 100-400 mA cm⁻². As shown in Fig. R17, in the neutral electrolyte, the CO₂ fraction increased significantly with current density (~6.1% to ~23.3%), consistent with reported severe carbonate crossover. To exclude contributions from HCO₃⁻ decomposition in the electrolyte itself, we conducted control tests using 1 M KHCO₃ without CO₂ feed, which showed only trace CO₂ signals (0.767-0.918% across 100-400 mA cm⁻²). In contrast, under acidic conditions, the CO₂ fraction remained much lower (~1.2% to ~12.4%). The suppression originates from the proton-enriched acidic diffusion layer, which swiftly converts carbonate and bicarbonate back into CO₂ near the cathode. This localized reconversion effectively hinders their crossover through the membrane, thereby reducing CO₂ leakage to the anode.

Although the AEM-based acidic MEA allows some carbonate transport, the crossover is reduced by about half relative to neutral operation. This trend is consistent with the expected suppression of carbon loss in acid-fed systems. Importantly, our system achieved a peak single-pass carbon efficiency (SPCE) of ~16% towards C₂H₄ at 10 A when reducing the CO₂ feed rate to 60-100 sccm (Revised Manuscript Fig. 3d). This value compares favorably with recent large-scale benchmarks, such as Xu et al., where at a total current of 10 A the DAT electrode (81 cm²) obtained a FE_{CH₄} of 54% and a single-pass carbon efficiency towards CH₄

of nearly 10%.^[1] Together, these results provide quantitative evidence that our acid-fed MEA sustains competitive carbon efficiency under industrially relevant conditions, and that CO₂ crossover does not compromise our main conclusions.

[ref. 1] Xu, Z., Lu, R., Lin, Z.-Y., Wu, W., Tsai, H.-J., Lu, Q. et al. Electrocatalytic CO₂ reduction to methane at ampere-level current density in an acid-fed MEA. *Nat. Energy*. **9**, 1397-1406 (2024).

Fig. R17 | CO₂ crossover analysis in different MEA electrolytes at current densities of 100-400 mA cm⁻². (a) Neutral MEA using 1 M KHCO₃ with AEM. The CO₂ fraction in the anode

outlet gas increased from ~6.1% to ~23.3% with rising current density, evidencing severe carbonate crossover. (b) Acidic MEA using 0.5 M $\text{K}_2\text{SO}_4/\text{H}_2\text{SO}_4$ with AEM. The CO_2 fraction remained much lower, ranging from ~1.2% to ~12.4% across the same current range. (c) Direct comparison of CO_2 crossover between neutral (1 M KHCO_3) and acidic (0.5 M $\text{K}_2\text{SO}_4/\text{H}_2\text{SO}_4$) electrolytes, highlighting the suppression of CO_2 crossover under acidic conditions.

No.	Current density/ mA cm^{-2}	$\text{CO}_2/\%$	$\text{O}_2/\%$	$\text{N}_2/\%$
1	100	0.767	3.909	120.964
2	200	0.830	6.334	118.344
3	300	0.918	9.297	114.697
4	400	0.829	10.223	112.993

Table R1 | Control experiment of anode outlet gas composition in 1 M KHCO_3 electrolyte without CO_2 feed at different current densities (100-400 mA cm^{-2}). The measured CO_2 fraction remained at trace levels (0.767-0.918%) across all current densities, confirming that the CO_2 detected in neutral MEA operation originates primarily from carbonate crossover rather than direct HCO_3^- decomposition in the electrolyte. N_2 (5 sccm) was supplied as protective carrier gas during measurements, and values are reported as volume percentages (%).

No.	Current density/ mA cm ⁻²	CO _{2, total} /%	CO _{2, crossover} /%	O ₂ /%	N ₂ /%
1	blank	-	-	0.119	126.907
2	100	6.834	6.067	4.956	111.955
3	100	9.250	8.483	4.793	109.110
4	200	13.945	13.115	7.529	98.377
5	200	17.329	16.499	8.741	90.837
6	300	22.301	21.383	11.962	77.538
7	300	22.223	21.305	11.962	77.581
8	400	24.171	23.342	14.872	69.627
9	400	23.010	22.181	15.231	70.946

Table R2 | Gas composition at the anode outlet of a neutral MEA system (1 M KHCO₃, pH ~7) at different current densities (100-400 mA cm⁻²). Outlet gas was analyzed after 1 h of electrolysis with two sampling injections, using N₂ as the carrier gas (5 sccm). All values are reported as volume percentages (%). The CO₂ fraction increases markedly with current density (from ~6.8% to ~24.2%), indicating significant CO₂ crossover due to carbonate migration across the AEM and subsequent protonation at the anode.

No.	Current density/ mA cm ⁻²	CO ₂ /%	O ₂ /%	N ₂ /%
1	blank	-	0.119	127.345
2	100	1.956	5.746	116.414
3	100	1.236	5.302	117.991
4	200	6.096	7.908	109.016
5	200	5.705	7.088	110.708
6	300	10.721	12.861	93.862
7	300	10.721	11.431	96.703
8	400	12.393	13.594	89.895

9	400	15.746	14.866	89.895
---	-----	--------	--------	--------

Table R3 | Gas composition at the anode outlet of an acidic MEA system (0.5 M K₂SO₄/H₂SO₄, pH ~1) at different current densities (100-400 mA cm⁻²). Outlet gas was analyzed after 1 h of electrolysis with two sampling injections, using N₂ as the carrier gas (5 sccm). All values are reported as volume percentages (%). The CO₂ fraction is much lower (from ~1.2% to ~12.4%) compared to the neutral case, reflecting suppression of CO₂ crossover by the acidic environment, although partial carbonate transport through the AEM still occurs.

Question 4: *The CNCP system shows a quite high cell voltage (4.2-5.0 V at 200 mA cm⁻²) and only moderate C₂H₄ selectivity (~50%). These values appear less competitive when compared to the best flow-cell configurations that are between 3-4V with higher FE.*

Response: We sincerely thank the reviewer for this important observation. The reported voltage range of 4.2-5.0 V corresponds to the scale-up 100 cm² MEA operated at 10 A. In addition, as shown in **Fig. R18** and **Revised Supplementary Fig. 12**, small-area MEA tests (1 × 1 cm²) exhibited markedly lower voltages of 2.6-4.0 V across 100-600 mA cm⁻², with ~2.8-3.3 V at 200 mA cm⁻². Notably, this value is lower than the 3.5 V reported for an acidic flow cell under comparable conditions (3 M KCl, pH ~1),^[1] demonstrating that our MEA platform can achieve competitive operating voltages.

Fig. R18 | Voltage-time profiles of the small-area MEA under different current densities.

(a-f) Chronopotentiometry measurements of a $1 \times 1 \text{ cm}^2$ MEA operated at current densities of 100 (a), 200 (b), 300 (c), 400 (d), 500 (e), and 600 mA cm⁻² (f). The corresponding average voltages are ~2.6, ~2.8, ~3.3, ~3.5, ~3.8, and ~4.0 V, respectively. Stable operation across the

tested current range highlights the ability of the acidic MEA configuration to maintain competitive voltages compared to reported acidic flow-cell systems.

We acknowledge that the voltage increases in scale-up devices, primarily due to the anodic oxygen evolution reaction (OER) on the commercial IrO₂ catalyst (i.e., Dimensionally Stable Anode). **Fig. R19** shows that at 200 mA cm⁻² the OER overpotentials were 3.2 V in 1 M KOH, 4.58 V in 1 M KHCO₃, and 3.66 V in 0.5 M K₂SO₄/H₂SO₄, each increasing further after 1 h of operation. These results highlight the intrinsic kinetic limitations of OER in acidic and neutral electrolytes, which are exacerbated in large-area MEAs. We agree that reducing the cell voltage is essential, and we believe that rational design or replacement of the anode catalyst could substantially lower the overpotential and enhance overall energy efficiency in future work.^[2] Importantly, although our system does not yet reach the lowest voltages reported in optimized flow cells, it represents, to our knowledge, the first demonstration of acidic MEA operation at industrially relevant 10 A with >300 h stability and ~50% C₂H₄ Faradaic efficiency. We respectfully suggest that these findings establish a complementary direction to flow-cell research, particularly by addressing scale-up durability in acidic CO₂-to-C₂ electrolysis.

Fig. R19 | Oxygen evolution reaction (OER) polarization curves of commercial IrO₂ anode catalysts in different electrolytes. (a-c) Linear sweep voltammetry (LSV) of commercial IrO₂ electrodes measured before and after 1 h OER operation in (a) 1 M KOH (pH 14), (b) 1 M KHCO₃ (pH 7.9), and (c) 0.5 M K₂SO₄/H₂SO₄ (pH 1.9). At 200 mA cm⁻², the overpotentials were 3.2 V in 1 M KOH, 4.58 V in 1 M KHCO₃, and 3.66 V in 0.5 M K₂SO₄/H₂SO₄, which further increased to 3.45, 4.70, and 3.76 V, respectively, after 1 h operation. These results highlight the kinetic limitations of commercial IrO₂ catalysts under neutral and acidic conditions, contributing to the higher voltages observed in large-area acidic MEAs.

[Ref. 1] Y. Zhao, L. Hao, A. Ozden et al. Conversion of CO₂ to multicarbon products in strong acid by controlling the catalyst microenvironment. *Nat. Synth.* **2**, 403-412 (2023).

[Ref. 2] X. Wang, P. Li, J. Tam et al. Efficient CO and acrolein co-production via paired electrolysis. *Nat. Sustain.* **7**, 931-937 (2024).

Question 5: *The long-term stability claims may be overstated: "... a full-cell voltage of ~4.5 V under a total current of 10 A (current density of 204 mA cm⁻²) achieved a Faradaic efficiency of 50% for CO₂-to-C₂H₄ conversion with long-term stability over 300 h." Yet in Fig. 3e, FE drops below 50% well before 100 h, and the voltage increases to 5 V around 100 h.*

Response: We sincerely thank the reviewer for this careful observation. As correctly noted, Fig. 3e shows that the Faradaic efficiency for C₂H₄ decreases below 50% after ~100 h, while the cell voltage gradually rises to ~5 V. Our original phrasing of "long-term stability over 300 h" was intended to emphasize that the electrolyzer remained continuously operational at 10

A for more than 300 h without sudden failure, with C₂H₄ production sustained throughout the test (**data shown in Revised Supplementary Table 1**).

We fully acknowledge that the FE and voltage trends reflect a gradual performance decline during extended operation. This behavior is mainly associated with Cu₃N-to-Cu transformation and GDL wetting, which we have explained in greater detail in our response to Question 6 and further clarified in the Revised Manuscript. Nevertheless, the COF-modified system significantly outperformed the unmodified Cu₃N electrodes, which lost stability within ~30 h under identical conditions (**Revised Supplementary Fig. 15**), thereby underscoring the stabilizing effect of the COF.

To prevent any possible overstatement, we have revised the text in manuscript to specify that the system demonstrates >300 h of continuous operational durability at 10 A, while the ~50% FE benchmark is maintained primarily within the first ~100 h. This modification offers a more accurate and balanced description of the stability profile.

Question 6: *The reasons for the improved stability are not clearly explained, and the explanation for the performance decay isn't convincing. According to Fig. 3e, both FE and voltage decline steadily at a similar rate over the 300h test, which doesn't quite match the typical behavior of GDL flooding.*

Response: We sincerely thank the reviewer for this constructive comment. The enhanced durability of the CNCP electrode is attributed to the Th-TF COF interlayer, which enriches K⁺ and accelerates OH⁻ transport, thereby suppressing HER and carbonate precipitation. This

dual ion-regulation effect, verified by in-situ XRFs, Raman, FTIR, and MD simulations, explains the markedly improved stability compared with the rapid degradation of bare Cu_3N .

To clarify the gradual performance decay, we first examined structural and chemical changes using XRD, operando XANES, and Raman spectroscopy. As shown in **Fig. R20a**, XRD shows that the characteristic reflections of Cu_3N at $\sim 23.3^\circ$, 40.9° , and 47.6° vanish after long-term operation, while a broad peak at $\sim 43\text{-}45^\circ$ appears, consistent with metallic Cu (111), confirming a phase transformation from Cu_3N to Cu. In **Fig. R20b**, operando XANES further reveals that the post-reaction Cu K-edge XANES spectrum closely resembles metallic Cu foil, with the edge position ($E_0 \approx 8979\text{-}8981$ eV) matching Cu^0 , a weakened and broadened white line near ~ 8995 eV, and diminished oscillations indicative of reduced coordination and nano crystallinity. In contrast, in **Fig. R20c and d**, Raman spectra of the Th-TF COF collected before and after 300 h electrolysis display nearly identical profiles, with preserved C=N (~ 1620 cm^{-1}), aromatic C-C (~ 1570 cm^{-1}), and N-H (~ 1344 cm^{-1}) vibrations, while the persistence of the weak keto C=O band and absence of new absorptions exclude oxidative degradation of the COF framework.

Fig. R20 | Structural evolution of Cu_3N catalyst before and after CO_2RR . (a) XRD patterns of pristine Cu_3N with reflections at $\sim 23.3^\circ$, 40.9° , and 47.6° (PDF#86-2284), which vanish after CO_2RR , while a broad peak at $\sim 43\text{--}45^\circ$ emerges from metallic Cu (111); peaks at $\sim 25^\circ$ and $\sim 55^\circ$ arise from the carbon paper. (b) Post-reaction Cu_3N shows a Cu K-edge XANES nearly identical to metallic Cu, with the edge aligned to Cu foil and a weakened white line, evidencing reconstruction into Cu^0 with minimal Cu-N coordination. (c) Raman spectra of Cu_3N nanoparticles, CNCP before, and after reaction, showing disappearance of the Cu-N band ($\sim 500\text{--}650\text{ cm}^{-1}$) after CO_2RR . (d) Raman spectra of Th-TF COF nanoparticles, CNCP before, and after reaction, showing preserved bands at $\sim 1620\text{ cm}^{-1}$ (C=N), $\sim 1570\text{ cm}^{-1}$ (C=C), and $\sim 1344\text{ cm}^{-1}$ (N-H), confirming COF stability during electrolysis.

We further conducted complementary TEM/EDS and SEM-EDS analyses provide morphological evidence for progressive degradation. As shown in **Fig. R21**, time-dependent TEM figures reveal that pristine Cu_3N features an ordered Cu-N lattice with uniform Cu and N distribution, but after 100 h the surface becomes roughened and porous with weakened N signals, and after 300 h the N signal is almost absent, indicating nitrogen depletion and reorganization into Cu-rich phases. In **Figs. R21e and g**, cross-sectional SEM-EDS shows that the GDL maintains porosity and localized K and S signals near the catalyst layer at 100 h, but by 300 h the structure becomes denser with electrolyte signals penetrating deep into the GDL, providing direct evidence of gradual infiltration and loss of hydrophobicity of GDL. Collectively, these results indicate that the steady performance decay arises from two coupled processes— Cu_3N -to-Cu transformation and GDL wetting—rather than catastrophic flooding, while the Th-TF COF interlayer itself remains chemically intact and functionally active.

Fig. R21 | Long-term stability and structural evolution of CNCP electrodes during CO_2 electroreduction. (a) Stability test of the CNCP electrode in 0.5 M K_2SO_4/H_2SO_4 at a constant current of 10 A (49 cm^2 GDE), showing cell voltage and C_2H_4 Faradaic efficiency over 300 h.

(b) STEM-EDS mapping of pristine Cu_3N , confirming uniform Cu and N distribution. Scale bars: 20 nm. (c) Atomic-resolution ADF- and ABF-STEM images of pristine Cu_3N along the [001] zone axis, resolving the ordered Cu and N sublattices with an interplanar spacing of 3.5 Å. Scale bars: 20 nm. (d,f) TEM and EDS mapping after 100 h and 300 h electrolysis, respectively, showing progressive nitrogen depletion and reconstruction into Cu-rich phases. Scale bars in (d): 50 nm. Scale bars in (f): 20 nm. (e,g) Cross-sectional SEM-EDS after 100 h and 300 h, respectively. At 100 h, the GDL retains porosity with K and S signals localized near the catalyst layer, whereas after 300 h extensive electrolyte infiltration and loss of porosity are evident, with K and S penetrating deep into the GDL. Scale bars: 200 μm .

Question 7: *There are errors that affect readability: In Figs. 2d and 3b, the yellow portion of the product distribution is undefined. Reference labels in Fig. 3f are wrong (pointing to the wrong references). In Fig. 4a, the unit for potassium concentration is missing.*

Response: We sincerely thank the reviewer for carefully identifying these mistakes, which indeed affected the readability of the initial submission. In the revised manuscript, we have corrected all of them. Specifically, in **Revised Manuscript Figs. 2d and 3b**, the yellow and green portions of the product distribution are now clearly defined as $\text{C}_3\text{H}_7\text{OH}$. The incorrect reference labels in Fig. 3f have been fixed to correspond to the correct literature sources listed in **Revised Manuscript**. In **Revised Manuscript Fig. 4b**, the missing unit for potassium concentration has been added (M). We apologize for these oversights and emphasize that they only concern figure presentation and do not affect the validity of the results or the scientific conclusions.

Fig. R22 | Corrected product distribution and benchmarking. (a) CO₂ reduction product distribution of CNCP electrodes in the acid-fed MEA system at different current densities (100-600 mA cm⁻²) with clearly defined fractions: H₂ (gray), CO (purple), CH₄ (blue), C₂H₄ (red), C₂H₅OH (pink), and C₃H₇OH (yellow). (b) CO₂ reduction product distribution at 10 A in the 100 cm² scale-up MEA electrolyzer, with clearly color coding: H₂ (gray), CO (purple), CH₄ (blue), C₂H₄ (red), CH₃CHO (pink), C₂H₅OH (light purple) and C₃H₇OH (green). (c) In-situ XRFs measurement of local K⁺ concentration on Th-TF COF/Cu₃N electrode, with units (M) explicitly indicated. (d) Comparison of this work with previous studies on acidic CO₂-to-C₂⁺ electrosynthesis at similar pH conditions. Reference labels have been corrected to Revised Manuscript and correspond to Table R4.

No	Works	Devices	Selectivity	FE _{C₂H₄}	Stability current	Stability
7	Science 372, 1074-1078 (2021) ¹	flow	FE _{C₂⁺} ~50%	~20%	1.2 A	12h
20	Angew. Chem. Int. Ed. 62 (2023) ²	flow	FE _{C₂⁺} ~70%	~35%	0.05A	5h
9	Nature Catalysis 5, 564-570 (2022) ³	flow	FE _{C₂⁺} ~89%	~40%	0.5 A	4.5h
22	Nature Communications 15, 491 (2024) ⁴	flow	FE _{C₂⁺} ~87%	~40%	0.6A	10h
11	Nature Synthesis 2, 403-412 (2023) ⁵	flow	FE _{C₂⁺} ~75%	~40%	0.2 A	30h
25	Nature Communications 15, 4821 (2024) ⁶	flow	FE _{C₂⁺} ~86%	~40%	0.9A	40h
23	Nature Communications 14, 1298 (2023) ⁷	flow	FE _{C₂⁺} ~69%	~50%	0.15A	35h
13	Nature Communications 14, 2387 (2023) ⁸	flow	FE _{C₂H₄} ~53%	~53%	0.2 A	10h
24	Nature Nanotechnology 19, 311-318 (2024) ⁹	flow	FE _{C₂H₄} ~61%	~61%	0.3 A	12h
12	Nature Catalysis 6, 763-772 (2023) ¹⁰	flow	FE _{C₂⁺} ~80%	~40%	0.1 A	155h
	Our Work	MEA	FE_{C₂⁺}~83%	~53%	10 A scale-up	300h

Table R4 | Comparison of this work with previous studies on the acidic electrocatalytic CO₂ to multi-carbon products at a similar electrolyte pH.